# Sensitivity of thermodynamic profiles retrieved from ground-based microwave and infrared observations to additional input data from active remote sensing instruments and numerical weather prediction models

Laura Bianco[1,2], Bianca Adler[1,2], Ludovic Bariteau[2], Irina V. Djalalova[1,2],
Timothy Myers[1,2], Sergio Pezoa[1,2], David D. Turner[3], and James M. Wilczak[2]

[1] CIRES, University of Colorado, Boulder, CO, USA
[2] NOAA, Physical Sciences Laboratory, Boulder, CO, USA
[3] NOAA, Global System Laboratory, Boulder, CO, USA

*Correspondence to*: Laura Bianco (Laura.Bianco@noaa.gov)

**Abstract.** Accurate and continuous estimates of the thermodynamic structure of the lower atmosphere are highly beneficial to meteorological process understanding and its applications, such as weather forecasting. In this study, the Tropospheric Remotely Observed Profiling via Optimal Estimation (TROPoe) physical retrieval is used to retrieve temperature and humidity
profiles from various combinations of input data collected by passive and active remote sensing instruments, in-situ surface platforms, and numerical weather prediction models. Among the employed instruments are Microwave Radiometers (MWRs), Infrared Spectrometers (IRS), Radio Acoustic Sounding Systems (RASS), ceilometers, and surface sensors. TROPoe uses brightness temperatures and/or radiances from MWRs and IRSs, as well as other observational inputs (virtual temperature from the RASS, cloud base height from the ceilometer, pressure, temperature, and humidity from the surface sensors) in a
physical-iterative retrieval approach. This starts from a climatologically reasonable profile of temperature and water vapor, with the radiative transfer model iteratively adjusting the assumed temperature and humidity profiles until the derived brightness temperatures and radiances match those observed by the MWRs and/or IRSs instruments within a specified uncertainty, as well as within the uncertainties of the other observations, if used as input. In this study, due to the uniqueness of the dataset that includes all the above-mentioned sensors, TROPoe is tested with different observational input combinations,
some of which also include information higher than 4 km above ground level (agl) from the operational Rapid Refresh numerical weather prediction model. These temperature and humidity retrievals are assessed against independent collocated radiosonde profiles under non-cloudy conditions to assess the sensitivity of the TROPoe retrievals to different input combinations.

**1 Introduction**

Knowing the thermodynamic structure of the atmosphere in the lowest few kilometres is of great importance for many studies including pollutant dispersion, severe weather, fire weather, wind and solar energy generation, model verification and evaluation, and atmospheric process understanding in general. Over the years the most reliable information on the thermodynamic state of the atmosphere has been derived by radiosonde launches, with strengths in terms of accuracy and vertical resolution and limitations in terms of temporal and spatial availability, which are well known to the atmospheric

science community. During the most recent years, an additional concern on the possibility of relying on radiosonde launches for atmospheric studies has added to the rest: helium shortage. On 29 March 2022, the U.S. National Weather Service Headquarters in Silver Spring, MD, USA, issued the following statement: "Effective March 29 and until further notice, the National Weather Service is reducing the frequency of weather balloon launches at several upper air locations in the United States due to a global supply chain disruption of helium"

(https://www.weather.gov/bou/HeliumShortageandBalloonLaunches).

Of course, radiosonde launches are not the only option available to observe the thermodynamic state of the lower part of the atmosphere. Several ground-based sensors (including in-situ or remote, and active or passive sensors) are currently available and operational in many geographical locations. In-situ sensors only provide point measurements (except for aircraft-based observations that can produce moderately-dense vertical profiles, although only sporadically in time) but can be used as a great

addition to the observations obtained by ground-based remote sensors. Active and passive sensors each have their strengths and limitations (Djalalova et al., 2022; Turner and Löhnert, 2021), which will be further detailed in the next section of this manuscript.

During fall 2021–winter 2022 (from the middle of September 2021 to the middle of January 2022) a series of in-situ, active, and passive ground-based remote sensors were deployed at Platteville, Colorado (CO), in the United States (Lat: 40.18N, Lon:

104.73W, Alt: 1503 m above ground level, agl). Among these were two passive ground-based Microwave Radiometers (MWR; Radiometrics MP-3000A), two passive ground-based infrared spectrometers (IRS; Atmospheric Sounder Spectrometer by Infrared Spectral Technology, ASSIST; LR Tech Assist-II), and an active ground-based Radio Acoustic Sounding System (RASS), associated to a 449-MHz Radar Wind Profiler (RWP). Also, surface meteorological observations of atmospheric variables such as pressure, temperature, moisture, wind speed and direction, solar radiation, and precipitation were measured

on a 10-meter SurfMet tower station. Two ceilometers were also deployed at the site, but the dataset analysed in this study mostly covers non-cloudy conditions, so their observations are only used for some discussion presented in the Appendix and relative to days with cloudy conditions. Finally, a total of 15 Vaisala RS-41 radiosonde launches were performed during the observational period, to use for comparison.

A photo of the observational site is presented in Fig. 1.


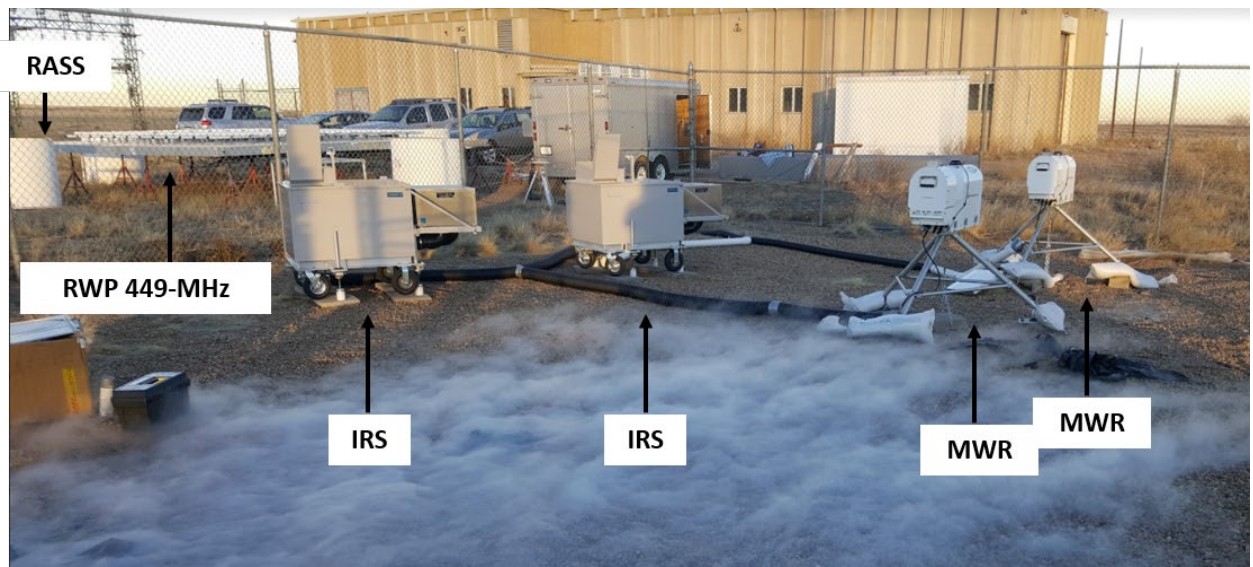

**Fig. 1. Photo of the instruments deployed at the Platteville, CO, site during the Platteville field campaign. Photo credit: Laura Bianco.**

A physical retrieval iterative approach can be used to retrieve thermodynamic vertical profiles from passive sensors, with the possibility of also including the information from other instruments, or from numerical weather prediction models. Other studies have compared MWR and IRS retrievals (Turner and Löhnert 2021, Turner and Blumberg 2019, Blumberg et al. 2015 and references therein) and some of the results presented here align with the previous findings. For example, we will show that the IRS clear-sky temperature retrievals have more independent pieces of information on both temperature and humidity profiles relative to the MWR retrievals. The uniqueness of the Platteville dataset is that it has collocated MWR, IRS, RASS, and surface sensors, which were combined in different configurations. The aim is to assess the sensitivity of the lower-atmospheric thermodynamic physical retrievals to a variety of input combinations, as well as the opportunity to assess the impact to the retrievals of including the input from a numerical weather prediction model.

## 2. Platteville dataset

### 2.1 Radiosonde observations

A total of 6 days with Vaisala RS-41 radiosonde launches are available: 3 days were in fall 2021 (27 September, 28 September, 5 October 2021) and 3 were in winter 2022 (22 December 2021; 10 January, 12 January 2022). When possible, the launches were scheduled at an interval of ~3 hours (e.g., at approximately 7 am, 10 am, 1 pm and 4 pm LT) to capture the daily evolution of the boundary layer development. To be able to track the evolution of the convective boundary layer, only clear-sky days

were chosen for this study. To compare with the retrieved thermodynamic profiles, each radiosonde profile is interpolated to
the vertical levels used in the physical retrieval iterative approach.

Information on the radiosonde launches, including temperature, T, and mixing ratio, MR, measured at surface and at around 5 km agl, and precipitable water vapor, PWV, computed as in Liu and Chen (2000), are summarized in Table 1.

| Radiosonde # | Day (mm/dd/yyyy) | Hour (UTC) | T at surf (°C) | T at 5 km agl (°C) | MR at surf (g kg$^{-1}$) | MR at 5 km agl (g kg$^{-1}$) | PWV (mm) |
|---|---|---|---|---|---|---|---|
| 1 | 9/27/2021 | 1407 | 16.7 | -13.6 | 4.3 | 0.7 | 12.6 |
| 2 | | 1711 | 26.9 | -14 | 3.3 | 0.7 | 12.4 |
| 3 | | 2004 | 30.6 | -13.7 | 3.5 | 0.6 | 13.1 |
| 4 | | 2315 | 30.3 | -13.9 | 3.2 | 0.5 | 12.5 |
| 5 | 9/28/2021 | 1334 | 14.2 | -14.6 | 5.9 | 1.5 | 19.8 |
| 6 | | 1700 | 26 | -13.8 | 5.5 | 0.9 | 18.1 |
| 7 | | 2009 | 27.7 | -14.6 | 3.6 | 1.5 | 14.6 |
| 8 | | 2302 | 25.7 | -15.1 | 4.1 | 1.6 | 16.5 |
| 9 | 10/5/2021 | 1700 | 21 | -11.7 | 3.7 | 0.2 | 12.2 |
| 10 | | 2000 | 27.1 | -11 | 3.4 | 0.7 | 11.4 |
| 11 | 12/22/2021 | 2017 | 15.5 | -19.6 | 0.4 | 0 | 3.6 |
| 12 | 1/10/2022 | 1802 | 3.1 | -21.3 | 2.2 | 0.2 | 3.5 |
| 13 | | 2104 | 7.2 | -20.9 | 2.8 | 0.2 | 3.4 |
| 14 | 1/12/2022 | 1822 | 5.9 | -18.8 | 3.2 | 0.6 | 8.3 |
| 15 | | 2108 | 13.4 | -18.6 | 2.6 | 0.5 | 8.6 |

**Table 1. Radiosonde launches available for this study (UTC = LT+6 in fall and LT+7 in winter).**


## 2.2 Ground-based remote sensors, MWR, IRS, and RASS: Strengths and weaknesses

MWRs and IRSs are passive sensors, with very sensitive receivers designed to measure the natural thermal emission from the earth's atmosphere. Microwave emissions in the water vapor (22-30 GHz) and oxygen (51-59 GHz) absorption bands can be used to retrieve vertical profiles of temperature and humidity from the MWR. The MWR used in this study has 35-channels in total (21 in the 22-30 GHz band, and 14 in the 51-59 GHz band). The MWR observed at the zenith and at 19.8º and 160.2º elevation angles on both sides of the zenith. We are aware that, when deployed in locations with unobstructed views, MWR's oblique scans can be performed down to 5º elevation angles and may provide better profile accuracy in the lowest 0–1 km agl layer (Crewell and Löhnert, 2007). Unfortunately, due to some obstructions, we could not go lower than 19.8º elevation angles. Associated noise levels were computed using the procedure described in Djalalova et al. (2022) and are listed in Table 2, averaged over the 3 days in fall (9/27-28 and 10/5/2021, for the MWR deployed in fall 2021) and over the 3 days in winter (12/22/2021 and 1/10-12/2022, for the MWR deployed in winter 2022). Additionally, in order to compute MWR's brightness temperature biases and correct for them before retrieving the thermodynamic profiles, we used the method referred to as "TROPoe BC" in Djalalova et al. (2022).

| Spectral bands | MWR Channels (GHz) | Noise levels (K) averaged over 3 days in fall | Noise levels (K) averaged over 3 days in winter |
|---|---|---|---|
| K-band | 22 | 0.4191 | 0.2805 |
| | 22.234 | 0.4417 | 0.2579 |
| | 22.5 | 0.4485 | 0.2740 |
| | 23 | 0.3708 | 0.2755 |
| | 23.034 | 0.3870 | 0.2776 |
| | 23.5 | 0.3937 | 0.2914 |
| | 23.834 | 0.3418 | 0.2584 |
| | 24 | 0.3476 | 0.2615 |
| | 24.5 | 0.3343 | 0.2634 |
| | 25 | 0.3219 | 0.2540 |
| | 25.5 | 0.2911 | 0.2962 |
| | 26 | 0.3265 | 0.2573 |
| | 26.234 | 0.2939 | 0.2428 |
| | 26.5 | 0.2940 | 0.2494 |
| | 27 | 0.3325 | 0.2252 |
| | 27.5 | 0.3271 | 0.2166 |
| | 28 | 0.3341 | 0.2093 |

| | | | |
|---|---|---|---|
| | 28.5 | 0.3123 | 0.2095 |
| | 29 | 0.2943 | 0.2714 |
| | 29.5 | 0.3288 | 0.3966 |
| | 30 | 0.3049 | 0.3216 |
| V-band | 51.248 | 0.4444 | 0.2475 |
| | 51.76 | 0.3958 | 0.2835 |
| | 52.28 | 0.4056 | 0.2572 |
| | 52.804 | 0.4098 | 0.2612 |
| | 53.336 | 0.3972 | 0.2742 |
| | 53.848 | 0.4133 | 0.2871 |
| | 54.4 | 0.3863 | 0.3352 |
| | 54.94 | 0.3698 | 0.3434 |
| | 55.5 | 0.5030 | 0.3269 |
| | 56.02 | 0.4478 | 0.3290 |
| | 56.66 | 0.4066 | 0.3526 |
| | 57.288 | 0.5614 | 0.4172 |
| | 57.964 | 0.4835 | 0.3940 |
| | 57.964 | 0.4506 | 0.4110 |

**Table 2. MWR channels and relative noise levels.**

The IRS measures infrared radiance in the spectral range from 500 cm$^{-1}$ to approximately 3,000-5,000 cm$^{-1}$ (depending on the system), with a spectral sampling of ~0.5 cm$^{-1}$ (Knuteson et al, 2004a, b). Spectral bands used for temperature and humidity retrievals are chosen from those sensitive to $CO_2$ (to retrieve temperature) and $H_2O$ (to retrieve water vapor). Spectral bands used for the IRS are: 538-588, 612-618, 624-660, 674-713, 713-722, 860.1-864.0, 872.2-877.5, 898.2-905.4 cm$^{-1}$.

Strengths of these passive instruments are their compact design, the relatively high temporal resolution (of the order of a few minutes, or less for the IRS), and the fact that they provide both temperature and moisture profile information and liquid water path. Conversely, a weakness of both the MWR and IRS are their rather coarse vertical resolution. Thermodynamic profiles can be retrieved from the MWR and IRS in both clear and cloudy conditions; however, because clouds are markedly more opaque in the infrared than the microwave, the IRS retrievals are more sensitive to errors in cloud base height (and thus require a collocated ceilometer measurement) and provide little-to-no information above the cloud (whereas the MWR retrievals provide some sensitivity above the cloud). Furthermore, the accuracy of the MWR-retrieved profiles is limited in the presence of rain (i.e., the retrievals often are satisfactory in light rain conditions as long as the radome is not wet) while the IRS does not sample the atmosphere in the presence of precipitation (a hatch at the top of the instrument automatically closes to protect

the optics if rain is detected by the instrument's surface sensor). Finally, while the IRS is a self-calibrating instrument, one of the weaknesses of the MWR is the need for non-trivial manual calibrations (Küchler et al., 2016, and references within). Prior to their use the MWRs were calibrated using an external liquid nitrogen target (Han and Westwater, 2000) and thoroughly
serviced (sensor cleaning, radome replacement, etc.). One IRS and one MWR were moved to another field campaign in mid-October 2021. These 2 units were used for the 9/27-28 and 10/5/2021 days runs and the other IRS and MWR were used for the 12/22/2021 and 1/10-12/2022 days runs.

Thermodynamic profiles from passive instruments such as MWRs and IRSs are often retrieved from the multi-wavelength brightness temperature or radiance observations using regression methods (linear, quadratic approaches), artificial intelligence
(neural networks), or physical iterative methods (Maahn et al 2020). In this study, we use a physical-iterative approach.

Being an active instrument, the RASS is more accurate and provides higher vertical resolution than passive instruments (Bianco et al, 2017). It emits a longitudinal acoustic wave in the vertical, causing a local compression and rarefaction of the ambient air. These density variations are tracked by the RWP associated with it, providing measurements of the speed of the propagating sound wave, which is proportional to the virtual temperature, Tv (North et al., 1973). Thus, RASSs are used to remotely
measure vertical profiles of virtual temperature in the boundary layer. For our dataset, the minimum RASS measurement is at 212 m agl, the maximum at 2228 m agl, and the vertical resolution is 106 m. The weaknesses of this instrument are the typically low temporal resolution (typically a 5 min averaged RASS profile is measured once or twice per hour), the altitude coverage limited to the lowest kilometres of the atmosphere (particularly in cooler and drier environments; May and Wilczak, 1993), and the fact that it only measures virtual temperature. Moreover, the maximum height reached by the RASS is variable and
limited by the advection of the propagating sound wave out of the radar's field of view (which can be different at different times of the day, as horizontal winds can have a strong diurnal cycle) and by sound attenuation (a function of both radar frequency and atmospheric conditions such as temperature, humidity; May and Wilczak, 1993). For example, during 2 of the days with available radiosonde measurements, the height coverage of the RASS was very different around the radiosonde time, as shown in Fig. 2 (panel a and b for 10 and 12 January 2022, respectively). The percentage of RASS data availability over
the ±30 minutes around all available radiosonde times are presented in Fig. 2c. Above 1.5 km agl the RASS data availability drops quickly to low values for this dataset, possibly due to the very dry atmospheric conditions experienced over the time period analysed here.

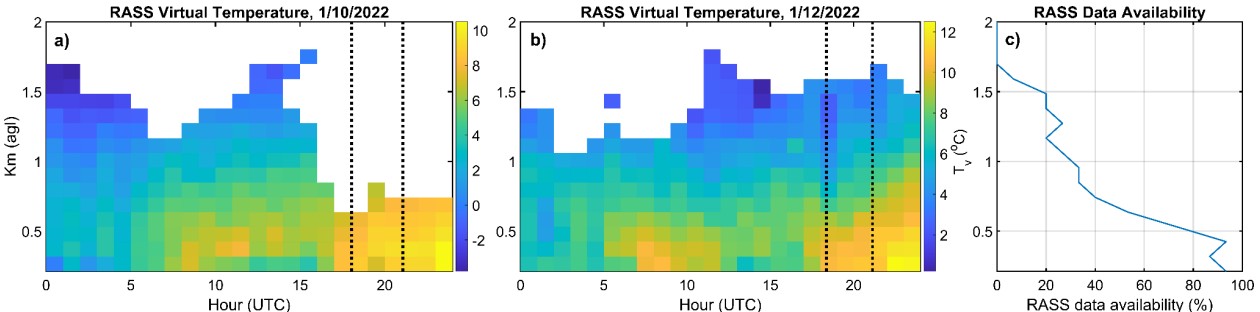

 **Fig. 2. Panels a and b: Time-height cross section of virtual temperature as measured by the RASS for 10 and 12 January 2022, respectively. Vertical dashed lines denote the radiosonde launch times. Panel c: Percentage of RASS data availability over ±30 minutes around all available radiosonde times.**

### 2.3 10-m SurfMet tower

The surface-meteorology instruments deployed on the 10-m tower include a propeller-and-vane anemometer and radiometer, and temperature, relative humidity, and barometric pressure sensors at 2-m. These provide a measure of a variety of quantities near the earth's surface, such as mean pressure, temperature, moisture, wind speed and direction, downwelling solar radiation, and precipitation. The surface observations of temperature and humidity (from a Campbell Scientific Model HMP45C temperature and relative humidity probe) and pressure are used in this study to constrain the retrieved thermodynamic profiles obtained by the physical retrieval iterative approach closer to the surface.

### 2.4 Ceilometer

The ceilometer deployed at Platteville is a Vaisala CL31 model, able to measure the height of the cloud base, leveraging pulsed diode lidar technology and single lens optics. The CL31 is engineered to deliver accurate data on multiple cloud layers even when conditions limit physical visibility. The CL31 model detects three cloud layers simultaneously to a range of 7.6 km.

### 2.5 The Rapid Refresh Numerical Weather Prediction Model

The Rapid Refresh (RAP) is the continental-scale, hourly-updated, assimilation/modelling system developed at the Global System Laboratory (GSL) of the National Oceanic and Atmospheric Administration (NOAA), and operational at the National Center for Environmental Prediction (NCEP). It has a 13-km horizontal grid spacing (Benjamin et al. 2016). Hourly thermodynamic vertical profile outputs of the operational RAP, extracted at the grid point closest to the location of the Platteville site, are used in the current study as a constraint to the upper level of the atmosphere (above 4 km agl) in some of the configurations of the physical retrieved iterative approach described in Section 3.

While other studies (Hewison 2007, Cimini et al 2015, Martinet et al 2020), employ an alternative approach with NWP model used directly within the a priori profile, in our study, we use the RAP as part of the observation vector. The uncertainty profiles for the RAP temperature and water vapor profiles are computed as the standard deviation over the surrounding neighboring grid points in the model. We additionally inflate the uncertainty of the RAP profiles by a factor of 3 for water vapor, while 1.5 °C is added to the temperature uncertainty to assure that the retrieval has the flexibility to consider the observations (if they diverge from the NWP).

## 3. Physical retrieval iterative approach

A physical retrieval iterative approach can be used to retrieve vertical profiles of thermodynamic properties from passive sensors, such as MWRs and IRSs. Other inputs, such as in-situ surface observations, and other ground-based observations, such as RASS (Djalaova et al., 2022), or water vapor differential absorption lidars (DIALs; Turner and Löhnert, 2021) can be included in a synergistic manner (Maahn et al., 2020). In this study, the Tropospheric Remotely Observed Profiling via Optimal Estimation (TROPoe) retrieval algorithm (formerly known as AERIoe; Turner and Löhnert, 2014; Turner and Blumberg, 2019; Turner and Löhnert, 2021) is employed. TROPoe's details are well presented in the references listed before and additional modification and improvements, that will be employed in future studies, are presented in (Adler et al., 2024). Its main characteristic is being an optimal estimation-based physical retrieval, initialized with a climatologically reasonable profile of temperature and water vapor. The mean state vector of the climatological estimates (prior) is a key component in the TROPoe framework, providing the level-to-level covariance needed to constrain the retrieval to realistic solutions. For this study the prior is calculated independently for each month of the year from 10 years of climatological radio sounding profiles in the Denver, CO, area. The radiative transfer models, MonoRTM (for the MWR; Clough et al., 2005) and LBLRTM (for the IRS; Clough and Iacono, 1995 and Clough et al., 2005), are iteratively repeated until the computed radiances match those observed by the MWR or IRS within the uncertainty of the observed radiances (and the uncertainties of the RASS virtual temperatures, if this is used as input) (Rodgers, 2000; Turner and Löhnert, 2014; Cimini et al., 2018; Maahn et al., 2020).

Due to the different instruments available at the Platteville site, TROPoe could be tested using different combinations of inputs to evaluate their impact on the retrievals in terms of information content, vertical resolution, and errors in temperature and mixing ratio profiles. The total number of TROPoe configurations tested is 12. The various TROPoe configurations investigated in the present study, and their reference numbers are summarized in Table 3.

| TROPoe Configurations | | | | |
|---|---|---|---|---|
| | | MWR | IRS | MWR+IRS |
| | None | #1 | #5 | #9 |
| Additional Inputs | RASS | #2 | #6 | #10 |
| | RAP (>4 km) | #3 | #7 | #11 |
| | RASS+RAP (>4 km) | #4 | #8 | #12 |

**Table 3. Configurations of observational and model inputs and their reference numbers for TROPoe physical retrievals investigated in the present study.**

We note that since the RASS measures virtual temperature, when this is included as input, the virtual temperature is computed at the end of each TROPoe iteration from the state vector and compared to the RASS measured virtual temperature.

All of the 12 TROPoe configurations also included in-situ measurements of temperature, pressure and humidity collected at the surface.

## 4. Results

In this section, the statistical performances of the various TROPoe configuration runs are assessed compared to the radiosonde launches available at the Platteville site. The time-height cross section of temperature derived by TROPoe including the IRS and surface observations only (TROPoe configuration #5) are presented for 28 September 2021, in the upper panel of Fig. 3 (panel a). Radiosondes launch times are denoted by the vertical dashed lines. The daily evolution of the temperature field is characterized by a decrease from the previous afternoon into the night time hours, the establishment of a temperature inversion close to the surface during the night time hours (between ~0300 – ~1300 UTC), and then the erosion of the temperature inversion starting at ~1400 UTC due to the surface being warmed by solar radiation. Sunrise for this day is 1255 UTC. The establishment of the convective boundary layer is then well visible during the day time hours (starting from ~1500 UTC).

A comparison of observed radiosonde and TROPoe retrieved profiles averaged ±30 minutes around the radiosonde times are shown in the middle and bottom panels of Fig. 3, for temperature (panels b, c, d, and e) and mixing ratio (f, g, h, and i), respectively. One of the advantages of TROPoe is that it provides in output the error covariance of the solution (Masiello et al, 2012). The square roots of the diagonal of this matrix provide the 1-sigma uncertainty profiles for temperature and humidity retrieved profiles (Turner and Löhnert, 2014). The 1-sigma uncertainty is represented with the shaded red areas in panels (b-i). Another important output of TROPoe is the averaging kernel matrix (Rodgers, 2000). The rows of this matrix provide the smoothing functions that could be applied to the radiosonde profiles when compared to the TROPoe retrievals, to minimize the fact that they have much higher vertical resolution than the retrieved profiles (Turner and Löhnert, 2014). While this would be appropriate, for this study the retrieved profiles will be compared to the unsmoothed radiosonde profiles because the averaging kernel matrix is different for the different TROPoe configurations and smoothing the radiosonde using different averaging kernel matrices would not provide a meaningful evaluation of the results.

The TROPoe profiles match qualitatively well the radiosondes, although the retrieved TROPoe profiles can miss some of the fine details in the surface temperature inversion detected by the radiosondes in the early morning hours (1334 UTC).

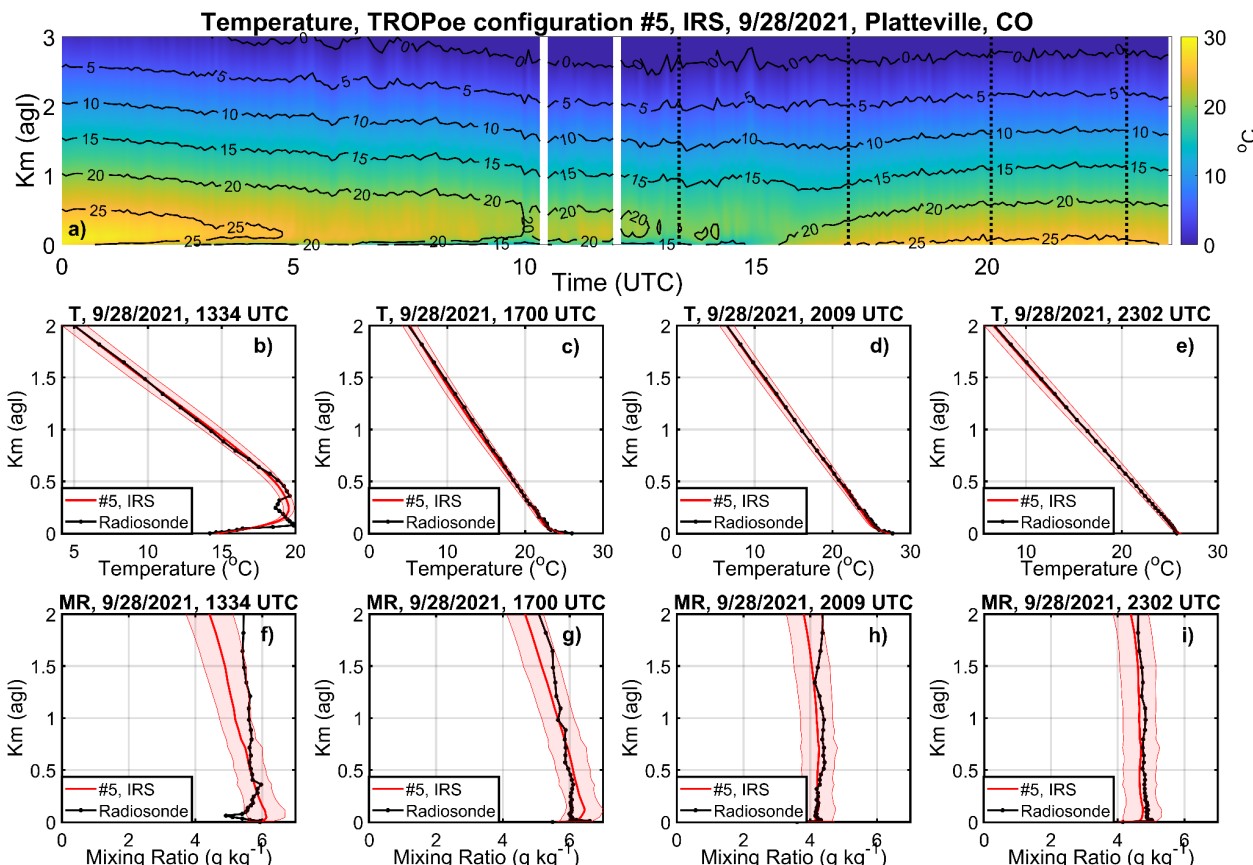

Fig. 3. Panel a: Time-height cross section of retrieved temperature for 28 September 2021 by the TROPoe run including the IRS and surface observations only. Vertical dashed lines denote the radiosonde launch times. Middle panels (b, c, d, and e): Temperature profiles as retrieved by the TROPoe run including the IRS and surface observations only (red) compared with radiosonde temperature observed profiles (black) at 1334, 1700, 2009, and 2302 UTC, respectively. Shaded areas indicate the 1-sigma uncertainty in the retrieved profiles. Bottom panels (f, g, h, and i): Same as in the middle panels, but for mixing ratio.

In the next section an analysis of the impact of including different inputs to the TROPoe approach in terms of degrees of freedom for signal and vertical resolution at each level of the retrievals will provide useful insights, before performing a quantitative statistical evaluation of the various thermodynamic retrieval configuration.

## 4.1 Analysis of physical retrieval characteristics

TROPoe not only provides output of the thermodynamic retrieved profiles, but also useful information about the calculated retrievals. The effective information content in any set of the analysed data is the degrees of freedom for signal (DFS; Cardinali, 2004). The cumulative DFS profile is a measure of the number of independent pieces of information in the observations below

the specified height (therefore, by definition, increases with height) and is also a TROPoe output. The DFS are, of course, dependent on the inputs used in TROPoe. Figure 4 presents the cumulative DFS as a function of height for each of the 12 TROPoe configurations tested in this study for temperature (panel a) and mixing ratio (panel b). Note that the vertical grid used in TROPoe is not uniform, with more frequent levels closer to the surface.

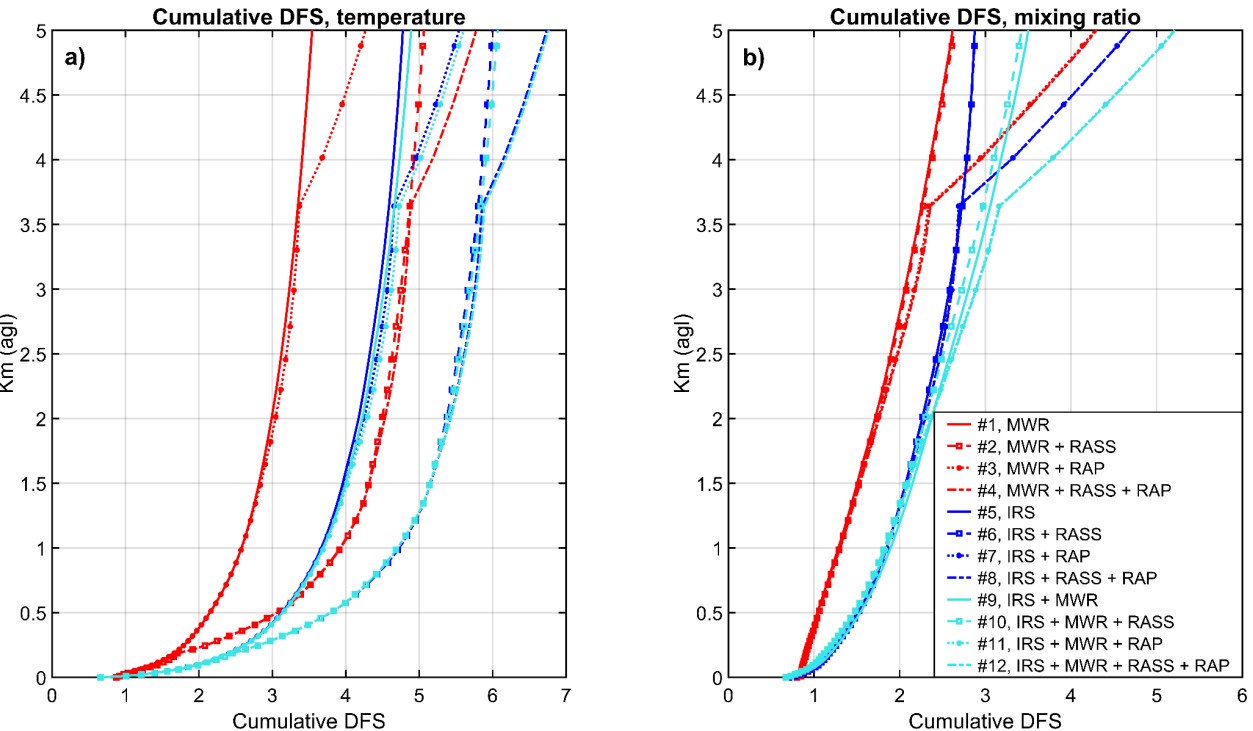

**Fig. 4. Panel a: Cumulative DFS for temperature as a function of height for each of the 12 TROPoe configurations. Panel b: Same as for panel a, but for mixing ratio.**

TROPoe configurations only including the passive instruments are presented with solid lines (configuration #1, MWR in red; configuration #5, IRS in blue; configuration #9, MWR+IRS in cyan). In Fig. 4a it is noticeable how the cumulative DFS for temperature for TROPoe configuration #1 is overall less than that of TROPoe configuration #5. For example, at 2 km agl, the cumulative DFS for configuration #1 is equal to 3, while for configuration #5 it is larger than 4. TROPoe configuration #9 only has slightly larger values for cumulative DFS for temperature compared to configuration #5, meaning that combining the MWR and IRS only adds a little bit of information to the IRS-only retrievals for these clear-sky cases. The TROPoe configurations including both the passive instruments and the RASS are presented with the dashed lines with open squares (configuration #2, MWR+RASS in red; configuration #6, IRS+RASS in blue; configuration #10, MWR+IRS+RASS in cyan). In these runs, the respective cumulative DFS for temperature increases substantially compared to the corresponding runs that do not include the RASS (configurations #1, 4, and 9, respectively). The impact of the RASS inclusion starts showing up from

the height of the first RASS measurement (212 m agl). It is noticeable how above 3 km agl the cumulative DFS stay pretty much constant for configurations #1, 2, 4, 5, 9, and 10, which means that above that height any additional information content is negligible. However, this is not the case when the RAP model is included to the TROPoe runs above 4 km agl. The cumulative DFS including both the passive instruments and the RAP are presented with the dotted lines with asterisks (configuration #3, MWR+RAP in red; configuration #7, IRS+RAP in blue; configuration #11, MWR+IRS+RAP in cyan). While the inclusion of the RAP to the passive instrument-only runs basically does not have an impact on cumulative DFS below 4 km agl, it is clearly important at this height and higher in the atmosphere, where an increase in cumulative DFS for temperature is visible comparing to the corresponding configurations that do not include the RAP (configurations #1, 5, and 9, respectively). Finally, when the RASS, the RAP, and the passive instruments are included in the TROPoe runs, the cumulative DFS are presented with the dash-dotted lines (configuration #4, MWR+RASS+RAP in red; configuration #8, IRS+RASS +RAP in blue; configuration #12, MWR+IRS+RASS+RAP in cyan). In these cases, the cumulative DFS for temperature are impacted by the inclusion of the RASS in the lower part of the atmosphere (from 212 m to around 2 km agl), and by the inclusion of the RAP in the upper part of the atmosphere (from 4 km agl and higher in the atmosphere), providing the highest values of cumulative DFS for all respective configurations that do not include both RASS and RAP. For example, configuration #1 (MWR only) has around 3.5 cumulative DFS for temperature at 5 km agl, while configuration #4 (MWR+RASS+RAP) has almost 6 cumulative DFS at the same height (similar impact is found comparing configuration #5, IRS only, to configuration #8, IRS+RASS+RAP, and comparing configuration #9, MWR+IRS, to configuration #12, MWR+IRS+RASS+RAP).

Fig. 4b shows the cumulative DFS for mixing ratio for the various TROPoe configurations. When using the passive instruments only as input in TROPoe, configuration #1 has again overall smaller cumulative DFS values than that of TROPoe configuration #5 and 9. For example, at 3 km agl, the cumulative DFS for configuration #1 is equal to approximately 2, while for configuration #5 the cumulative DFS is approximately 2.5 and for configuration #9 approximately 3. The TROPoe configurations including both the passive instruments and the RASS have similar values for cumulative DFS for mixing ratio compared to the respective runs not including the RASS. This is expected, as virtual temperature observations from the RASS are dominated by the ambient temperature (not moisture). Therefore, similarly to what was found by Djalalova et al. (2022), the RASS inclusion has little impact on the mixing ratio retrievals. On the contrary, when the RAP model is included in the TROPoe runs starting at 4 km agl, the cumulative DFS for mixing ratio including both the passive instruments and the RAP (configuration #3, MWR+RAP dotted red line with asterisks; configuration #7, IRS+RAP dotted blue line with asterisks; configuration #11, MWR+IRS+RAP dotted cyan line with asterisks) present larger values starting at 4 km agl and higher in the atmosphere. When the RASS, the RAP, and the passive instruments are included in the TROPoe runs (configuration #4, 8, and 12), the cumulative DFS for mixing ratio are presented with the dash-dotted lines and are very similar to the corresponding configuration runs with no RASS (configuration #3, 7, and 11, respectively). Finally, we note that the cumulative DFS for the TROPoe configurations that include both the passive MWR and IRS (cyan lines) are not very different compared to those only including the IRS (blue lines), except for cumulative DFS for mixing ratio above 2 km agl, where the inclusion of the MWR

in the TROPoe inputs results in increasing its values, for example at around 5 km the cumulative DFS for mixing ratio for configuration #5 is approximately 3, increasing to 3.5 for configuration #9. However, as mentioned before, the days included in this analysis are exclusively clear-sky, so this result could be different in the case of the presence of clouds.

As mentioned in the previous section, the rows of the averaging kernel provide a measure of the retrieval smoothing as a function of altitude, so the full width at half maximum of each averaging kernel row estimates the vertical resolution of the retrieved solution at each vertical level (Maddy and Barnet, 2008; Merrelli and Turner, 2012). Figure 5 presents the vertical resolution for each of the 12 TROPoe configurations tested in this study as a function of the height for temperature (panel a) and mixing ratio (panel b).

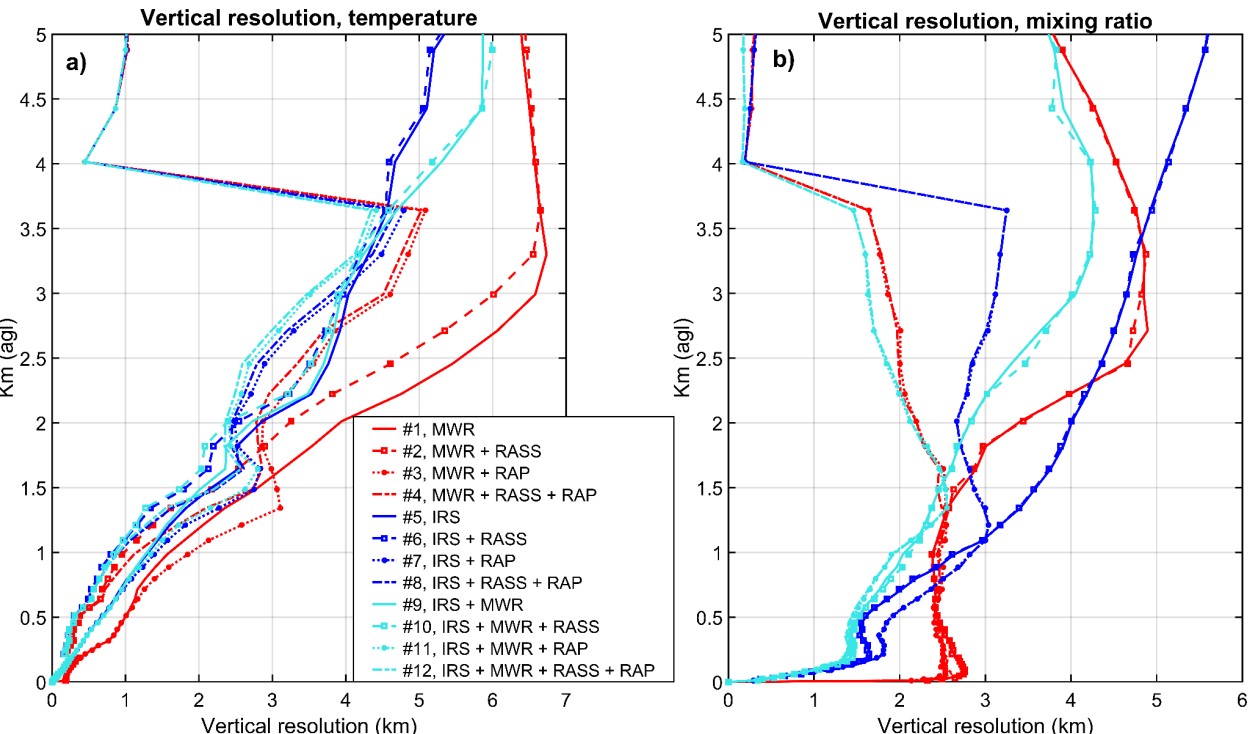

**Fig. 5. Panel a: vertical resolution of the retrieved temperature profiles as a function of the height for each of the 12 TROPoe configurations. Panel b: Same as for panel a, but for mixing ratio.**

For temperature, the vertical resolution of configuration #1 (MWR, solid red line) has the largest values compared to the other configurations. Configuration #5 (IRS, solid blue line) has better (i.e., smaller) vertical resolution for temperature, compared to configuration #1. When including the RASS together with the passive instruments to the TROPoe inputs (dashed lines with open squares: configuration #2, MWR+RASS in red; configuration #6, IRS+RASS in blue; configuration #10, MWR+IRS+RASS in cyan), the impact of the RASS is to improve (i.e., reduce) the vertical resolution values starting from the first height of RASS measurements (212 m agl) up to the maximum height where it provides measurements, but also

improves resolution in the layer above, up to 3.5 km agl. Above this height the RASS inclusion has no impact. This agrees with Djalalova et al. (2022), where it was found that the inclusion of the RASS improves the statistics also above the maximum RASS height.

When including both the RAP and the passive instruments to the TROPoe runs (dotted lines with asterisks: configuration #3, MWR+RAP in red; configuration #7, IRS+RAP in blue; configuration #11, MWR+IRS+RAP in cyan) the impact of the RAP

is to substantially improve (i.e., reduce) the vertical resolution values for temperature starting from 4 km up above in the atmosphere. For example, configuration #5 (IRS only) has a vertical resolution for temperature equal to around 5 km at 4.5 km agl, while configuration #7 (IRS+RAP) has a vertical resolution < 1 km at the same height (similar, if not larger impact is found comparing configuration #1, MWR only, to configuration #3, MWR+RAP, and comparing configuration #9, MWR+IRS, to configuration #11, MWR+IRS+RAP). This is not surprising as at this height most of the information content

comes from the RAP. The impact of the RAP on the vertical resolution for temperature is still visible below 4 km agl in the TROPoe run only including MWR as passive instrument (configuration #3, MWR+RAP, red dotted line with asterisks), but is negligible below that height for the other TROPoe runs. Finally, when including the RASS, the RAP, and the passive instruments in the TROPoe runs (dash-dotted lines: configuration #4, MWR+RASS+RAP in red; configuration #8, IRS+RASS+RAP in blue; configuration #12, MWR+IRS+RASS+RAP in cyan) the vertical resolution for temperature is

improved by the inclusion of the RASS in the lower part of the atmosphere and by the inclusion of the RAP in the upper part of the atmosphere, providing the best values of vertical resolution for temperature for all respective configurations that do not include both RASS and RAP.

Fig. 5b shows the vertical resolution for mixing ratio for the various TROPoe configurations. In this case there is again no impact with the inclusion of the RASS to the vertical resolution of mixing ratio for the various TROPoe configurations, but a

330 substantial impact (i.e., the improvement of the vertical resolution) when including the RAP as input to the TROPoe runs. In the case of mixing ratio, the best vertical resolution is obtained when using both passive instruments and the RAP.

## 4.2 Statistical analysis of physical retrieval profiles up to 5 km agl compared to radiosonde profiles

In this section, a quantitative statistical evaluation of the various thermodynamic retrieval configurations tested in this study is provided up to 5 km agl. The reason why this value for the maximum height is chosen is because the 0-5 km agl atmospheric

layer includes the surface and the boundary layer, as well as the 3.5-5.0 km transition layer where both the RAP and the observations make some contributions to the retrievals.

Figure 6 presents the mean absolute error (MAE) for each of the 12 TROPoe configurations tested in this study relative to the radiosonde observations as a function of the height for temperature (panel a) and mixing ratio (panel b).

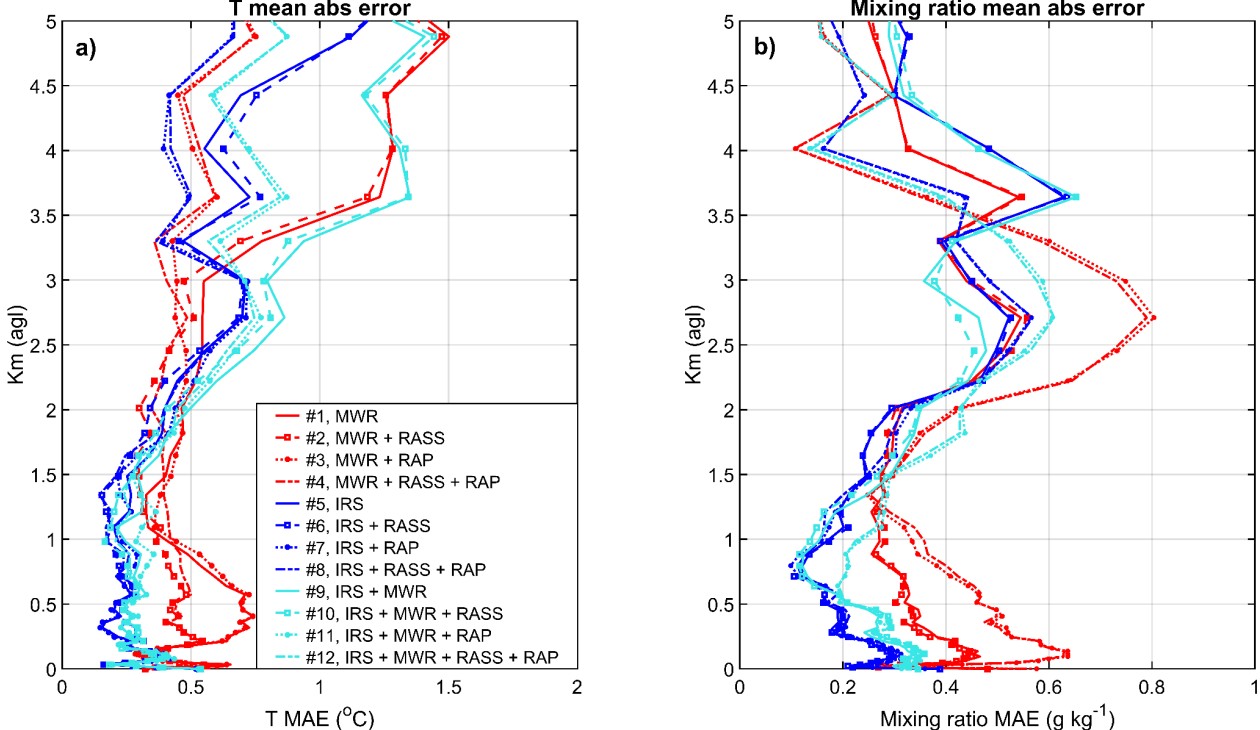

**Fig. 6. Panel a: Mean absolute error of the retrieved temperature profiles as a function of the height for each of the 12 TROPoe configurations. Panel b: Same as for panel a, but for mixing ratio.**

For temperature (Fig. 6a), the impact generated when including the RASS is to reduce MAE in the lower part of the atmosphere. This impact is larger for the TROPoe configurations including the MWR as the only passive instrument, as the initial MAE for this configuration (configuration #1, red solid line) is larger compared to the other configurations in the 0–1 km agl atmospheric layer. This is consistent with what was found in Bianco et al. (2017), that the MWR can struggle to get the details of the surface temperature inversions often observed at night or early morning hours. For configurations #6 and 10 the inclusion of the RASS is nevertheless still positive (i.e., the MAE is reduced). Also, for configurations #1, 2, and 10, it is again noticeable how the inclusion of the RASS improves the statistics also above the maximum RASS height, in agreement with Djalalova et al. (2022) and with Figs. 4a and 5a. While the impact of the RASS inclusion fades with height, the RAP inclusion provides a beneficial impact higher up in the atmosphere, for all configurations including it. Finally, when including both the RASS and the RAP, the MAE of all configurations is the best compared to the respective ones that do not include them. It is also noticeable that the inclusion of both MWR and IRS in the TROPoe inputs might not necessarily provide a better agreement with the radiosondes compared to the individual passive instruments used as input alone. This might be due to the fact that TROPoe will have to balance the information from the two passive instruments. Nevertheless, the combination of the two passive instrument (configuration #9) is still beneficial in the lower part of the atmosphere (< 2 km agl), where the MWR tents to struggle identifying the correct height and shape of inversions (configuration #9 better than configuration #1). Additionally,

the inclusion of both passive instruments in the TROPoe inputs might reveal a beneficial impact (see further discussion in the
Appendix) when cloudy conditions (over which radiosondes are not available for this dataset) are analysed.

For the mixing ratio (Fig. 6b), the impact of the RASS is almost negligible, as already expected from the considerations made in Section 4.1 and also in agreement with what was found in Djalalova et al. (2022). The impact of the RAP inclusion to the MAE of mixing ratio is in general positive to all configurations around and above the height of the RAP inclusion (4km agl). Nevertheless, the inclusion of the RAP generates a negative impact to the mixing ratio MAE below that height for the TROPoe
runs including the MWR as the only passive instrument. The different impact of the RAP inclusion in configuration #3 (MWR+RAP) compared to that in configuration #7 (IRS+RAP) might be due to the fact that the IRS has more information content in humidity than the MWR alone, so, for configuration #7 the retrievals below 4 km are better constrained by the observation. For configuration #3, differences between the NWP model and the observations above 4 km agl (NWP model being drier than the MWR observations) might spread (to counterbalance) in the lower part of the atmosphere.

In Fig. 7, the MAE and bias of temperature retrieved profiles compared to radiosondes (a and b panels, respectively) are averaged over the lowest 3 and 5 km agl (dashed and solid lines, respectively). The bias is computed as TROPoe temperature retrievals minus radiosonde temperature profiles. These averages are weighted over the vertical heights up to 3 and 5 km agl because, as mentioned above, the vertical grid used in TROPoe is not uniform, with more frequent levels closer to the surface. In this way equal height intervals contribute equally to the MAE and bias. The average up to 3 km agl will show more the
impact of the RASS inclusion to the MAE and bias values in a layer where the IRS and MWR have most of their information (Fig. 4), while the average up to 5 km agl will show more that of the RAP inclusion.

The MAE for temperature averaged over the lowest 5 km agl presents smaller values for configuration #5 (IRS only), compared to configurations #1 and 9 (MWR and MRW+IRS). All 3 of these configurations show some improvement by the inclusion of the RASS, but more so by the inclusion of the RAP. The inclusion of both the RASS and RAP shows similar values for MAE
of temperature to the runs with the RAP only included with the passive instruments. When averaging up to 3 km agl the values of MAE for temperature of configuration #1, 5, and 9 are smaller than the averages up to 5 km, as in general the MAE increases with height (Fig. 6a). When including the RASS in the TROPoe runs the impact is to further decrease the MAE averaged up to 3 km agl. The RAP inclusion does not show any impact on the temperature MAE when averaging over the lowest 3 km agl, as expected since the RAP is included only at 4 km agl and higher in the atmosphere. Overall, the MAE for temperature is
relatively small (~ 0.5 ºC) for all TROPoe configurations including the RASS, the RAP, and the passive instruments.

For the biases in temperature, all TROPoe runs have a slightly cold bias, which is improved (i.e., reduced) by the inclusion of the RAP when averaging over the lowest 5 km agl, and not impacted much by the inclusion of the RASS (except for the inclusion of the RASS in TROPoe runs using the MWR as the passive instrument). The inclusion of the RAP has a little impact on the temperature bias, when averaged over the lowest 3 km agl (making the bias slightly colder). The bias is degraded for
the configuration IRS+MWR compared to that of the instruments used alone. In all runs, the bias is smaller when averaging over the lowest 3 km agl, instead of 5 km agl.

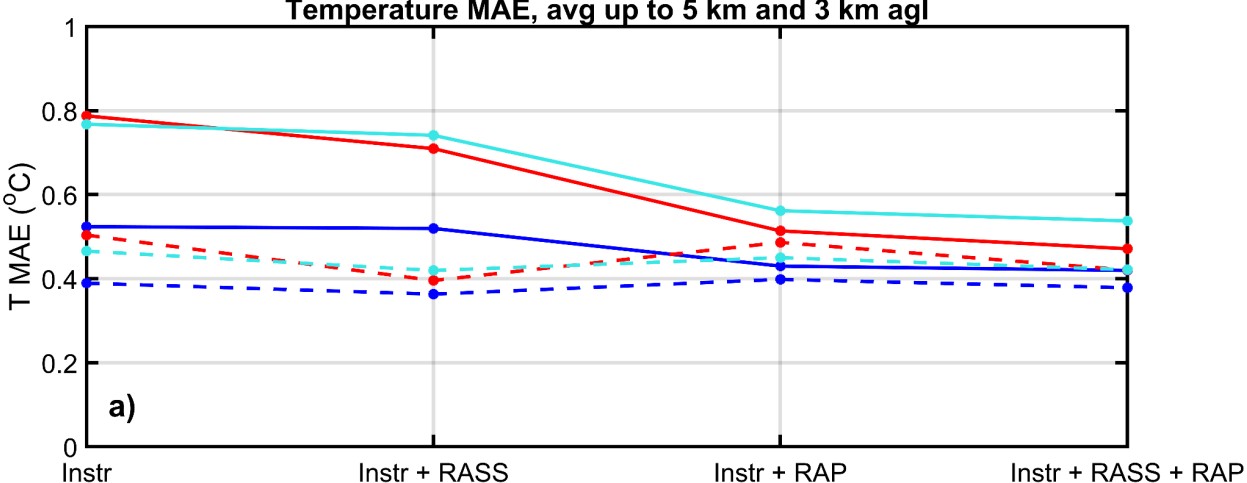

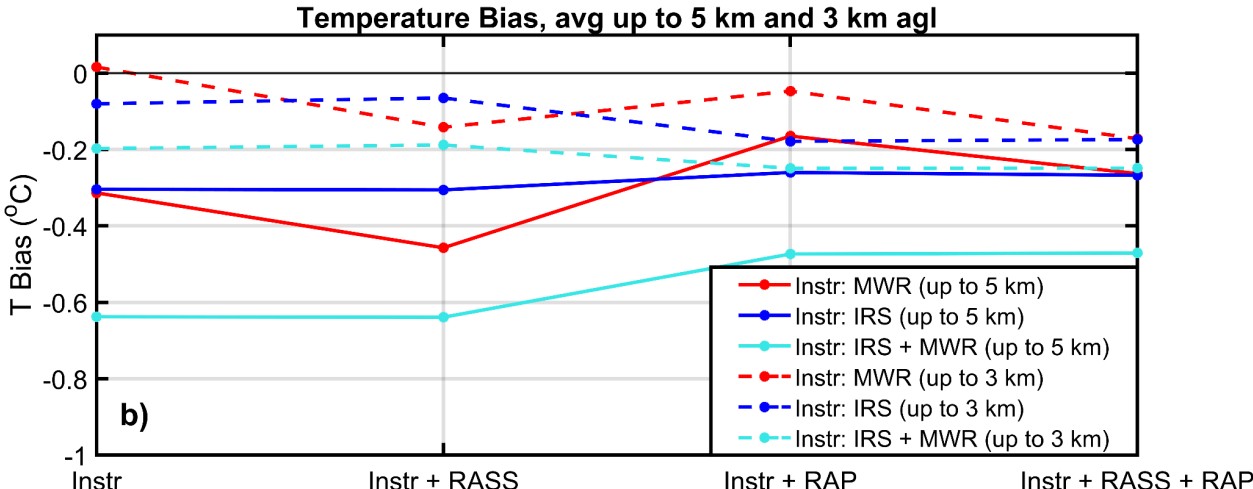

Fig. 7. Panel a: Mean absolute error of the retrieved temperature profiles averaged over the lower 3 (dashed lines) and 5 (solid lines) km agl for each of the 12 TROPoe configurations. Panel b: Same as for panel a, but for bias (TROPoe minus radiosonde temperature).

In Fig 8, a and b panels, MAE and biases are computed for the TROPoe retrieved profiles of mixing ratio, again averaged over the lowest 3 and 5 km agl (dashed and solid lines, respectively). The MAE for mixing ratio is relatively small for all TROPoe configurations ($< 0.5$ g kg$^{-1}$). As already noted from Fig. 6b, the effect of the inclusion of the RAP in the TROPoe runs that only include the MWR as the passive instrument is to slightly degrade the MAE for mixing ratio in the lower in the 5 km of the atmosphere and a little more in the lowest 3 km, but it slightly improves the MAE values for the TROPoe runs that only include the IRS as the passive instrument when averaging in the lower in the 5 km of the atmosphere.

Similarly to the impact on MAE, the RASS inclusion does not show any impact on the mixing ratio bias, while the impact of the RAP on the mixing ratio bias is again to slightly degrade it for the TROPoe runs that include the MWR in input.


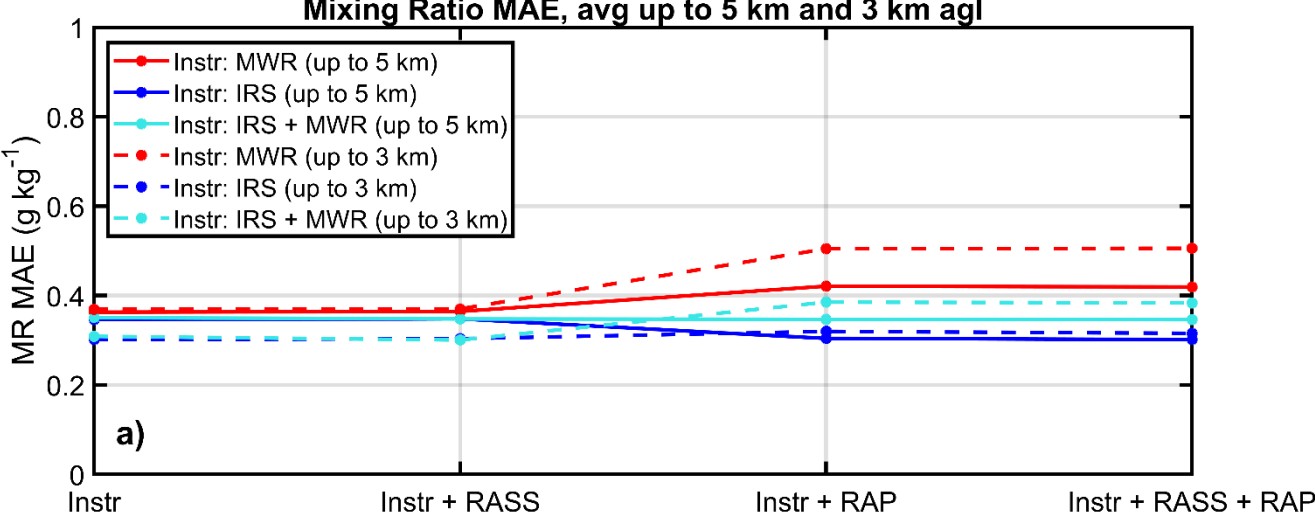

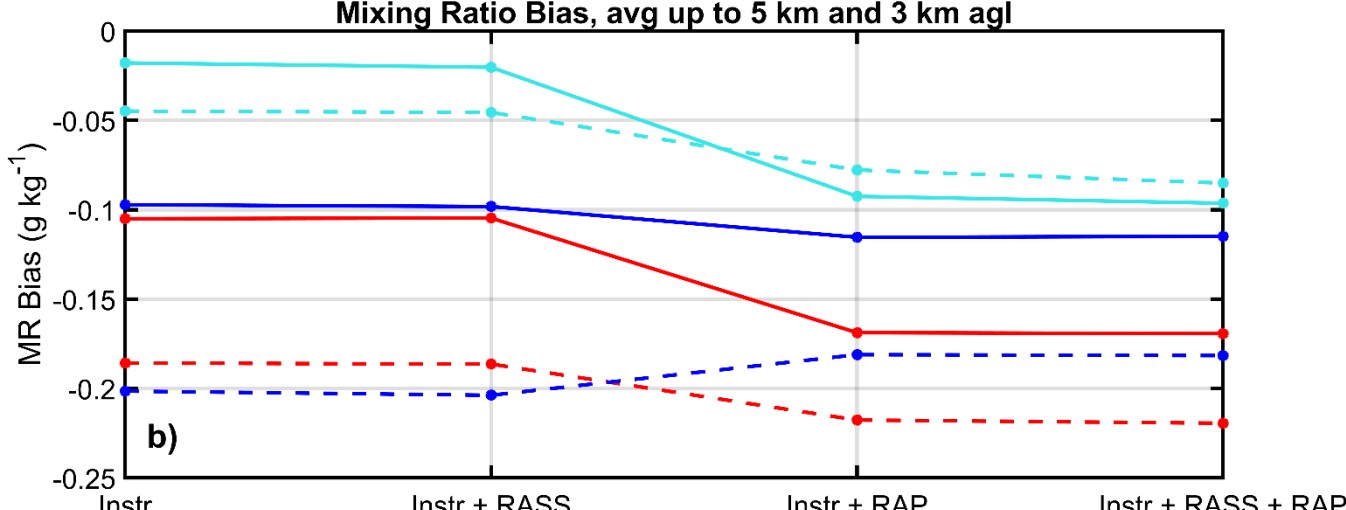

Fig. 8. Same as in Fig. 7, but for mixing ratio.

## 4.3 Statistical analysis of potential temperature lapse rate

In many applications there is the need for information derived from potential temperature profiles, for example to determine atmospheric stability and differentiate between stable and unstable conditions. Passive instruments usually tend to smooth the retrieved profiles, due to their coarser vertical resolution (Solheim et al., 1998; Reehorst, 2001). Nevertheless, Bianco et al. (2017) found a good agreement ($R^2 = 0.91$) in the values of potential temperature lapse rates ($d\Theta/dz$) derived from MWRs, compared to in-situ observations, particularly when the lapse rate is computed in the layer 50–300 m agl. Similarly, Klein et

al. (2015) found a value of $R^2 = 0.93$ (in fall) and $R^2 = 0.98$ (in summer) in the agreement between the ambient temperature lapse rates ($dT/dz$) derived from an IRS, compared to radiosonde observations, when the lapse rate is computed in the layer 10–100 m.

Here we investigate if and how the different combinations impact the potential temperature lapse rate in comparison to the radiosonde derived ones. Figure 9 presents scatter plot comparisons of potential temperature lapse rate over the 0–318 m agl

layer of the atmosphere from radiosondes and TROPoe retrievals for all of the 12 TROPoe configurations (panel a, including passive instruments only; panel b, including passive instruments and RASS; panels c, including passive instruments and RAP from 4 km agl; and panel d, including passive instruments, RASS, and RAP from 4 km agl), and corresponding best-fit lines. When the potential temperature lapse rate is computed over the 0–318 m agl layer of the atmosphere the agreement in terms of coefficient of determination between the TROPoe configurations and the radiosondes is impacted by the addition of the

RASS to the inputs, particularly from configuration #1 (MWR only, $R^2 = 0.9$) to configuration #2 (MWR+RASS, $R^2 = 0.98$). This drastic improvement over configuration #1 is mainly caused by an underestimation of very stable lapse rates and an overestimation of slightly unstable lapse rates by the MWR retrieval. The coefficient of determination for potential temperature lapse rate over the 0–318 m agl layer is higher for configuration #5 (IRS only, $R^2 = 0.97$), compared to configuration #1 (MWR only), also in terms of best-fit line (Fig. 9a). When including the RASS a small improvement occurs from configuration #5

(IRS only, $R^2 = 0.97$) to configuration #6 (IRS+RASS, $R^2 = 0.98$), and from configuration #9 (MWR+IRS, $R^2 = 0.97$) to configuration #10 (MWR+IRS+RASS, $R^2 = 0.98$).The inclusion of the RAP does not impact the agreement between the TROPoe configurations and the radiosondes (Fig. 9, panel c versus panel a; and Fig. 9, panel d versus panel b). The potential temperature lapse rate was also computed over different layers of the atmosphere, i.e., 0–95, 0–512, and 0–983 m agl. Statistical results of the comparisons relative to all layers of the atmosphere considered are reported in Table 4. Very similar values are

found for the coefficient of determination of ambient temperature lapse rates. The impact of the RASS and RAP addition to the TROPoe inputs does not change the potential temperature lapse rate statistic over the 0–95 m agl layer (as the first height of the RASS is above the 0–95 m agl and the RAP inclusion happens well above all these selected layers), nor in the 0–983 m agl layer as the values of the coefficient of determination are already very high, with not much room for improvement. Clearly, the inclusion of the RASS is positive in the 0–318 m agl layer, and the results show that the potential temperature lapse rates

determined by the TROPoe retrievals can be reliably used to determine the stability of the atmosphere, basically for all the configurations.

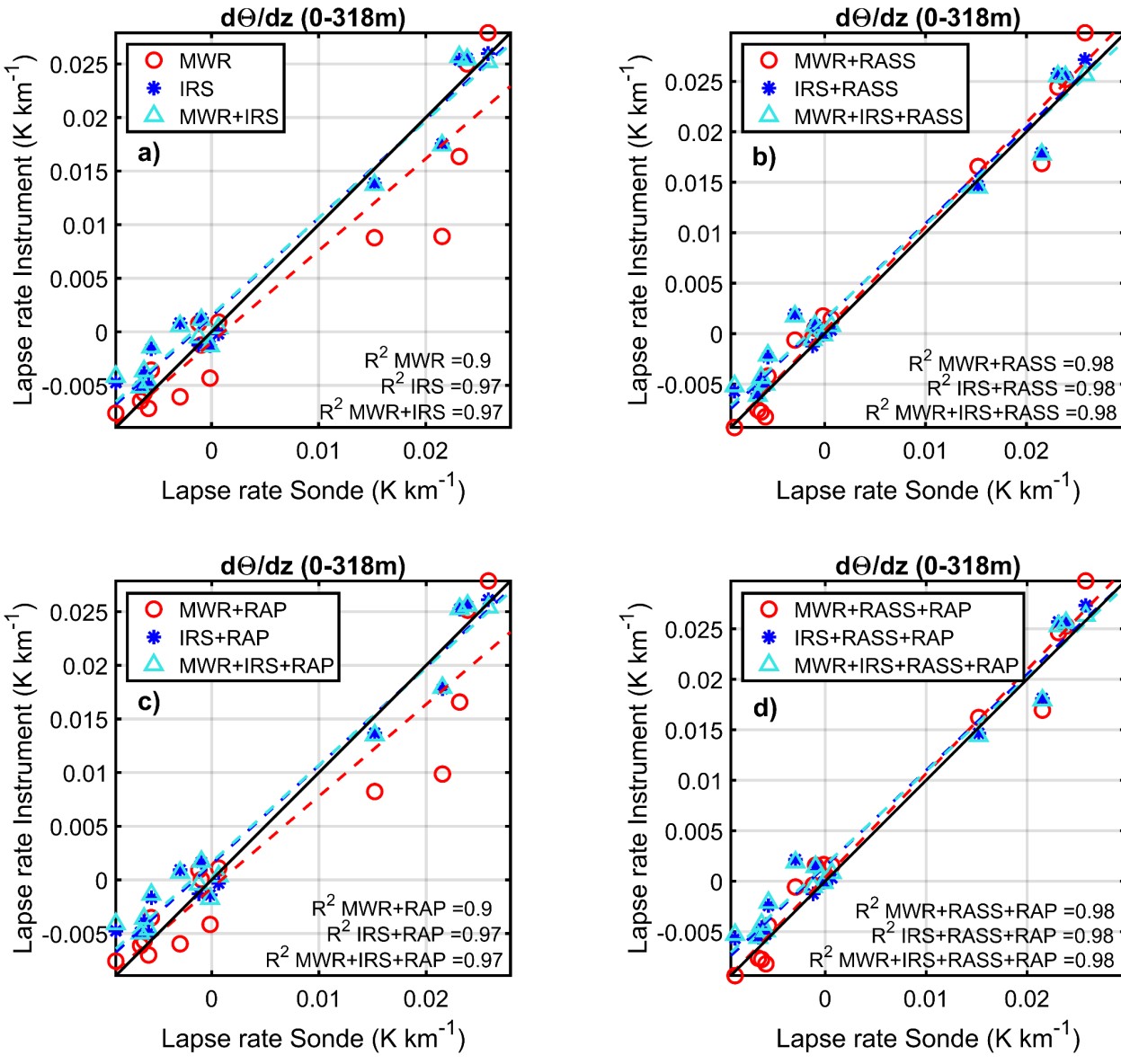

**Fig. 9.** One-to-one comparison of potential temperature lapse rate over the 0–318m agl layer of the atmosphere from radiosondes (x-axis) and TROPoe retrievals (y-axis) for TROPoe configurations #1, 5, and 9 (panel a), TROPoe configurations #2, 6, and 10 (panel b), TROPoe configurations #3, 7, and 11 (panel c), and TROPoe configurations #4, 8, and 12 (panel d). Dashed lines are best-fit lines.

| TROPoe | R2 |
|---|---|
|  |  |

| configuration | dΘ/dz<br>0–95 m agl layer | dΘ/dz<br>0–318 m agl layer | dΘ/dz<br>0–512 m agl layer | dΘ/dz<br>0–983 m agl layer |
|:---:|:---:|:---:|:---:|:---:|
| #1 | 0.93 | 0.9 | 0.94 | 0.99 |
| #2 | 0.93 | 0.98 | 0.99 | 0.98 |
| #3 | 0.93 | 0.9 | 0.94 | 0.99 |
| #4 | 0.93 | 0.98 | 0.99 | 0.98 |
| #5 | 0.9 | 0.97 | 0.99 | 0.98 |
| #6 | 0.89 | 0.98 | 0.99 | 0.98 |
| #7 | 0.9 | 0.97 | 0.99 | 0.98 |
| #8 | 0.89 | 0.98 | 0.99 | 0.99 |
| #9 | 0.89 | 0.97 | 0.99 | 0.98 |
| #10 | 0.89 | 0.98 | 0.99 | 0.98 |
| #11 | 0.89 | 0.97 | 0.99 | 0.98 |
| #12 | 0.89 | 0.98 | 0.99 | 0.99 |

**Table 4. Statistical comparisons in terms of coefficient of determination, between potential temperature and ambient temperature lapse rates for all 12 TROPoe configurations and radiosonde observations, over different layers of the atmosphere.**

## 5. Conclusions

In this study, the Tropospheric Remotely Observed Profiling via Optimal Estimation (TROPoe) physical retrieval is used to retrieve temperature and humidity profiles from various combinations of input data collected by passive (MWRs and IRSs)

and active (RASS) remote sensing instruments, in-situ surface platforms, and numerical weather prediction models (RAP) at a measurement site located in Platteville, Colorado, in the United States. TROPoe is tested with different observational input combinations, and assessed against collocated radiosonde profiles under non-cloudy conditions to identify optimal combinations. Results show that in non-cloudy conditions, when adding the RASS and RAP to the passive instruments in the TROPoe inputs, the statistical agreement with radiosondes is in general improved. The RASS and RAP have impact over

different layers of the atmosphere, as the RASS is mostly available in the lower part of the atmosphere, and the RAP is assimilated only higher than 4 km agl. Nevertheless, the improvement from the inclusion of both RASS and RAP is noticeable in terms of cumulative degrees of freedom for signal (DFS), vertical resolution, mean absolute error and bias, for temperature and humidity profiles; the impact of the RASS on humidity retrievals is negligible due to the nature of RASS measurements. For temperature, in agreement with Djalalova et al. (2022), it was found that the inclusion of the RASS improves the statistics

also above the maximum available RASS height. For all TROPoe configurations including both the RASS and the RAP, the MAE for temperature was found to be between ~0.4 – ~0.5 ºC (when averaged up to 3 and 5 km, respectively), and for mixing

ratio ~0.4 g kg$^{-1}$ in the dry environment experienced in this analysis. Results from this study also confirm that potential temperature lapse rates computed using TROPoe retrievals for any of the combinations can be used to assess the stability of the atmosphere and that the inclusion of the RASS to the TROPoe inputs can further improve the agreement with radiosonde estimates of lapse rate. Although for this dataset (clear-sky conditions) it is found that the inclusion of the combined MWR and IRS observations to the TROPoe inputs did not necessarily provide a better agreement with the radiosondes compared to the configurations using the individual passive instruments as input alone, we believe that this might be different when cloudy conditions will be analysed. For example, as mentioned above, since the IRS does not provide information above thick clouds because clouds are opaque to infrared transmission, we expect that the combination of MWR and IRS will have a larger impact and be more beneficial in cloudy conditions, not analysed in this study, as in that case the information retrieved by the MWR might supplement the lack of information above the cloud layer from the IRS (see further discussion in the Appendix).

The uniqueness of the Platteville, CO, dataset is in the availability of co-located IRS, MWR, RASS, ceilometer, and surface observations and RAP output. Nevertheless, the radiosonde sample size available for this study is relatively small and the days under analysis were clear-sky, so the results could be different in other climatological environments, which will be investigated in our future studies. The instruments deployed at the Platteville, CO, site were later moved to other sites for other field campaigns, and the continuation of the analysis presented here will include repeating the investigation over different geographical location and atmospheric conditions, when radiosonde launches will be available.

**Appendix**

Due to the lack of radiosonde during cloudy conditions, it is not possible to assess the performances of our different TROPoe configurations in cloudy conditions. Nevertheless, we recognize that even just looking at some of the TROPoe outputs, would provide insights of what benefits could be derived by the combination of the passive instruments in presence of clouds.

The overall conditions over the time period of our analysis were mostly clear skies. Nevertheless, we used the ceilometer observations and found 2 days that revealed the presence of clouds with corresponding relatively large values of liquid water path (LWP). Specifically, these 2 days are 9/30/2021 and 1/9/2022.

In Fig. 1A are presented the time series of cloud base height from the ceilometer (CBH, panels a and b) and liquid water path from the microwave radiometer (LWP, panels c and d), for 9/30/3021, and for 1/9/2022.

In panels c and d, all values of LWP are plotted in red in the time series, and values with LWP > 20 g kg$^{-1}$ are coloured in blue.


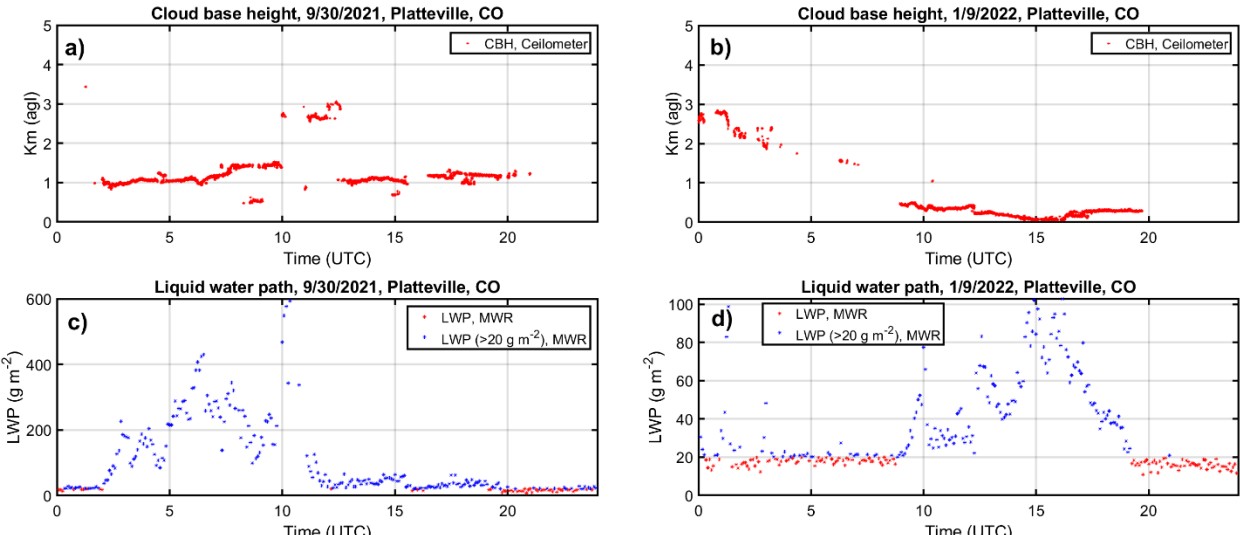

**Fig. 1A. Time series of CBH from the ceilometer (panels a and b) and LWP from the microwave radiometer (panels c and d), for 9/30/3021 (panels a and c), and for 1/9/2022 (panels b and d). In panels c and d, all values of LWP are plotted in red in the time series, and values with LWP > 20 g kg$^{-1}$ are coloured in blue.**


For these 2 days we ran TROPoe using as input the MWR only, the IRS only, and the combination of the MWR+IRS (similarly to the runs presented in the main body of the manuscript, in this exercise we also included in-situ measurements of temperature, pressure and humidity collected at the surface). These configurations were numbered in the main body of the manuscript as configurations #1, #5, and #9, respectively. Although we cannot assess which retrievals would agree better with an independent

observation (i.e. radiosonde), the time-height cross sections of temperature and mixing ratio for these TROPoe configuration runs (not shown) revealed the impact of the clouds in the retrievals relative to configuration #5 particularly, mostly in terms of time continuity above the cloud base height from previous profiles when we go from a situation with clouds observed to clear-sky periods, or clouds being detected at a higher altitude, or vice-versa. These discontinuities did not appear in the TROPoe retrievals of configuration #1, which was expected as stated before, as clouds are markedly opaquer in the infrared

than the microwave, and provide little-to-no information above the cloud (whereas the MWR retrievals provide some sensitivity above the cloud). Also, as speculated before, we noticed that the retrievals of configuration #9 (not shown) seems to benefit from the combination of MWR+IRS, as the information retrieved by the MWR supplements the lack of information above the cloud layer from the IRS. For configuration #9, the time discontinuities above cloud base height seen in the retrievals for configuration #5 are not present anymore.

This seems promising of the combination of the MWR and IRS in cloudy conditions.

For these 2 cloudy days, we averaged cumulative DFS and vertical resolution for temperature and mixing ratio, over the time periods with observed clouds and corresponding LWP > 20 g kg$^{-1}$.

In Fig. 2A, in dashed lines, these computed cumulative DFS as a function of the height are in the upper panels for configurations #1, #5, and #9 (temperature in panel a and mixing ratio in panel b) and in dashed lines vertical resolution as a function of the

height for configurations #1, #5, and #9 are in the lower panels (temperature in panel c and mixing ratio in panel d). In the same figure cumulative DFS and vertical resolutions obtained for the same configurations, but clear-sky days presented in the main body of the manuscript, are replotted for comparison, in solid lines.

From Fig. 2Aa we notice that the cumulative DFS for temperature for #1 (MWR only, red dashed line) in cloudy conditions do not show much difference with the profile of configuration #1 obtained for clear-sky days (red solid line). The cumulative

DFS for temperature in cloudy conditions relative to configuration #5 (IRS only, blue dashed line) do show a kink around 1 km agl, most likely due to the presence of clouds around that height. The cumulative DFS for temperature in cloudy conditions relative to configuration #9 (IRS + MRW, cyan dashed line) show an increase in respect to those of configuration #1 and #5. This increase is larger to that of configuration #9 in clear-sky days (cyan solid line, which showed little difference from that of configuration #5, blue solid line), which supports the speculation of the benefit that can be obtained by the combination of

MWR and IRS in cloudy conditions.

From Fig. 2Ab we notice that the cumulative DFS for mixing ratio in cloudy conditions for #1 (MWR only, red dashed line) again does not show much difference with the profile of configuration #1 relative to clear-sky days (MWR only, red solid line). The cumulative DFS for mixing ratio in cloudy conditions relative to configuration #5 (IRS only, blue dashed line) do show a large decrease compared to the profile of configuration #5 for clear-sky days (IRS only, blue solid line), most likely

due to the presence of clouds. However, the cumulative DFS for mixing ratio in cloudy conditions relative to configuration #9 (IRS + MWR, cyan dashed line) show an increase in respect to those of configuration #1 and #5 in the same cloudy conditions. This increase again supports the speculation of the benefit that can be obtained by the combination of MWR and IRS in cloudy conditions.

From Fig. 2Ac we notice a similar profile in cloudy conditions to that relative to clear-sky days for configuration #1 (MWR

only, red dashed and solid lines, respectively) for the vertical resolution of temperature, but a degradation in vertical resolution of temperature in cloudy conditions for the TROPoe configuration #5 (IRS only, blue dashed line), particularly in the lower part of the atmosphere, where clouds were detected. Combining the MRW and IRS in the TROPoe runs (configuration #9, cyan dashed line), derives in an improvement of vertical resolution of temperature in the lower part of the atmosphere particularly in cloudy conditions, where the vertical resolution of configuration #5 was degraded.

From Fig. 2Ad we notice a similar behaviour in the profiles of vertical resolution of mixing ratio, i.e.: similar to those obtained in clear-sky conditions for configuration #1 (MWR only, red lines), a degradation of it for configuration #5 (IRS only, blue lines) in the lower part of the atmosphere, and a benefit obtained by the combination of MWR and IRS in cloudy conditions (configuration #9, cyan dashed lines).

Although this analysis is relative to only 2 cloudy days, the results motivate us even more to investigate the benefit of the
combination of MWR and IRS during cloudy conditions on other datasets. As mentioned in the Conclusions, the instruments
deployed at the Platteville, CO, site were later moved to other sites for other field campaigns, and the analysis performed at
Platteville will be continued, when radiosonde launches will be available for comparison.

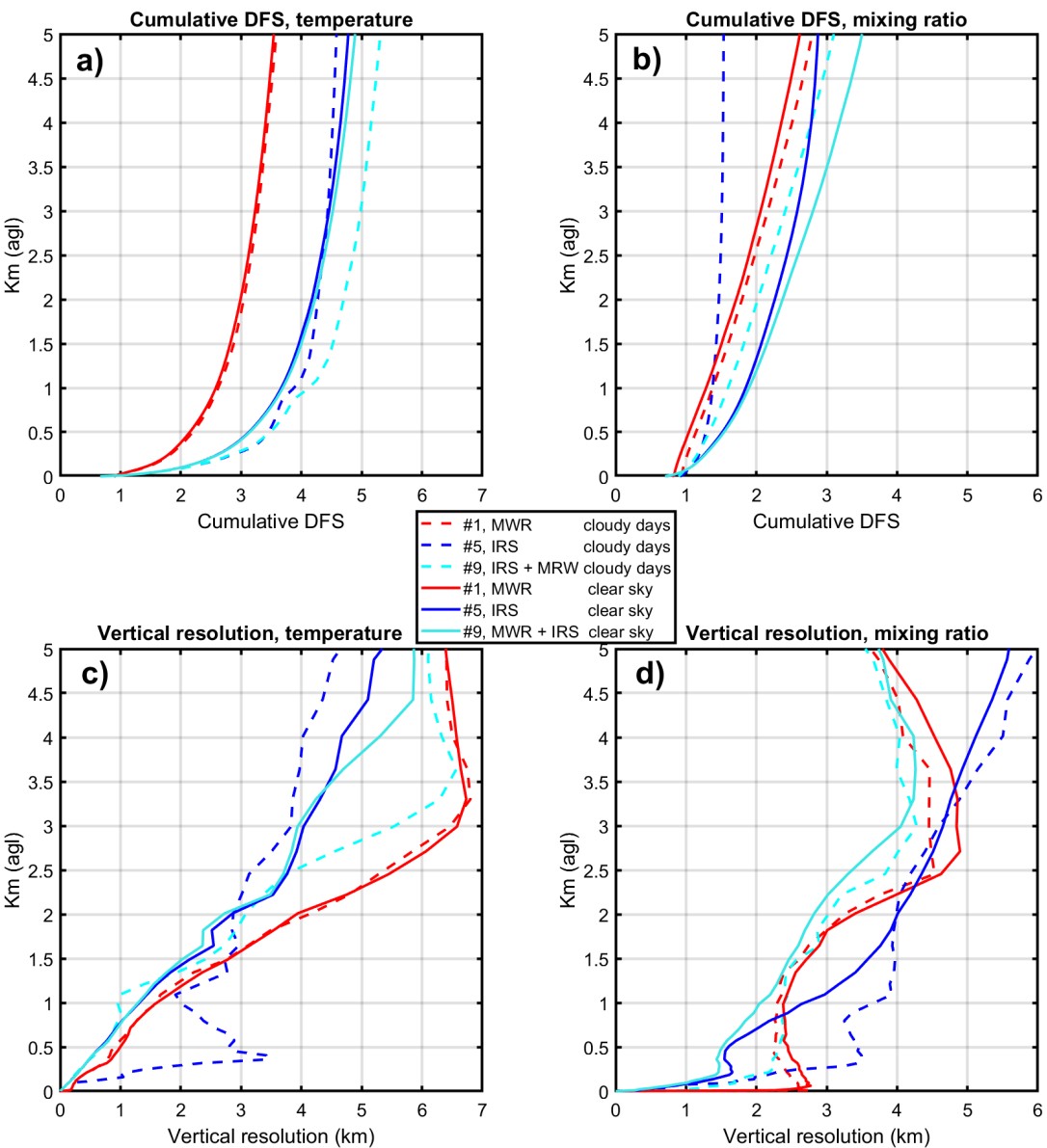

**Fig. 2A. Cumulative DFS for temperature (panel a) and mixing ratio (panel b) and vertical resolution for temperature (panel c) and**
**mixing ratio (panel d) of the retrieved TROPoe profiles as a function of the height for configurations #1, #5, and #9, for the cloudy**

days 9/30/3021 and 1/9/2022 (dashed lines). Same variables, for the same configurations, but for the clear-sky days presented in the main body of the manuscript are plotted in solid lines, for comparison.

### Code and data availability

The TROPoe code is available via DockerHub. Datasets used in this study are available from (Bianco, 2024).

### Acknowledgements

The authors wish to thank all engineers of the NOAA/ESRL/PSL group who take care of maintaining the operational functionality and availability of the data collected by the RASS and SurfMet stations located in Platteville, CO and for hosting additional instruments deployed for this study. Funding for this work was provided by the NOAA Atmospheric Science for Renewable Energy program. This research was supported by the NOAA cooperative agreement with CIRES, 570 NA22OAR4320151.

### Author contributions

All authors contributed to the instruments' deployment, calibration, and to the radiosonde launches. L Bianco completed the data analysis and prepared the manuscript with contributions from BA, L Bariteau, IVD, TM, SP, DDT, and JMW.

### Competing interests

One of the (co-)authors is a member of the editorial board of Atmospheric Measurement Techniques. The peer-review process was guided by an independent editor, and the authors also have no other competing interests to declare.

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
