# Peer review of "Sensitivity of thermodynamic profiles retrieved from ground-based microwave and infrared observations to additional input data from active remote sensing instruments and numerical weather prediction models"

_Atmospheric Measurement Techniques, 2023_

## Referee Comment (RC1)

**Sensitivity of thermodynamic profiles retrieved from ground-based
microwave and infrared observations to additional input data from active remote sensing
instruments and numerical weather prediction models**

**General comments :**

This paper evaluates the sensitivity of temperature and humidity retrievals to different combinations of ground-based remote sensing instruments as well as NWP model information. It uses the state of the art optimal estimation retrieval algorithm TROPOe that has been used in several scientific publications to demonstrate the improved accuracy of thermodynamic retrievals when combining together active remote sensing (water vapor lidar, RASS, ceilometers) and passive instruments (ground-based microwaver radiometers (MWR) and infrared spectrometers (IRS)). This new study is relevant for the scientific community due to the uniqueness of co-located remote sensing instruments at the same site (MWR, IRS, RASS, ceilometers and tower measurements) over a long time period (fall 2021-winter 2022). It also contributes to a new perspective of improvement for TROPOe by including NWP model information above 4 km altitude within the algorithm. The results show that RASS contributes to improved temperature retrievals within the boundary layer while the inclusion of NWP information significantly improves the temperature and humidity retrievals mainly above 3 km. Overall the manuscript is very well written, figures are well presented and explained. However, I am concerned by the conclusions of this study that are quite limited due to the small dataset of co-located radiosoundings available (only 15 RS). As clearly stated by the authors, conclusions taking into account cloudy-sky conditions could be different for the synergy between MWR and IRS observations for example. I am also wondering if degradations in the retrievals could happen when including the NWP profile above 4 km directly in the observation vector (if I understood well) that might not been observed in this study due to the limited dataset of validation. In fact, the retrieval algorithm could try to minimize the distance between the atmospheric state and the a priori as well as observations with potentially non-consistent observations (the NWP profile could potentially try to push the algorithm in a direction while the observation in another direction). The author should clarify this point and better justify the methodology used.

Additionally, I think a discussion on the differences in TROPOe results (DFS, vertical resolution, mean uncertainties, retrieved profiles) for clear-sky versus cloudy-sky days (as the observations are all available during one year) with respect to the different configurations could be beneficial to the paper. Even without available radio-soundings, DFS, vertical resolution and uncertainties could have been discussed as well statistical distributions of retrieval differences. If the TROPOe retrievals are available over a long time period and the authors can lead this analysis, I think it could improve the current manuscript.

Finally, the manuscript does not clearly state if MWRs are used with off-zenith scans. Several publications have demonstrated the significant increase in DFS by including off-zenith observations. If MWR observations have been used with zenith only observations, the comparison with the AERI instrument is not really fair and we could except more accurate temperature and lapse rate retrievals with the inclusion of off-zenith channels. This will not drastically change the

conclusion with RASS and RAP but it would definitely affect the conclusion comparing single passive instruments.

Afer taking into account the scientific points listed, I would recommend the publication of this manuscript in AMT.

**Major comments :**

- **Section 2.2** : As shown in Djalalova 2022 , even after nitrogen calibration, significant biases can be observed in MWR observations. Optimal estimation is very sensitive to biases in observations. Did you implement any bias correction or quality control of the brightness temperatures before applying TROPOe ? Even if the IRS is self-calibrating, was there any check on potential biases in IRS also ?

In line with the general evaluation : did you use off-zenith observations for the MWR ? If only zenith observations have been used, it should be clearly stated through all the manuscript that all conclusions comparing the MWR and IRS retrievals are underestimating the capability of current MWRs that are generally used with boundary layer scans to improve the vertical resolution of temperature profile. If another configuration with zenith and off-zenith observations could be included in the manuscript, it would be beneficial for the discussion.

**Section 2.4** :It is not clear to me if the RAP model is used within the a priori profile or within the observation vector. If it is used within the observation vector how was defined the corresponding observation error covariance matrix ? As mentioned in the overal evaluation, I am concerned that this methodology could degrade the retrievals in case of larger errors in the NWP profile non consistent with the other observations that might not been observed due to the limited number of radiosounding observations. Could you clearify and justify the methodology ? Several publications using an alternative approach with NWP model used directly within the a priori profile should also be cited (Hewison 2007, Cimini et al 2015, Martinet et al 2020) to discuss the difference with your methodology.

**Figures 4 and 5:**

- When the information content from observations is small, the inclusion of external information from NWP models has a significant impact on the retrievals. This is demonstrated in this study in figures 4 and 5 both for temperature above 4 km agl and to a larger extent for humidity above 1.5 km. Could the improvement on water vapor be larger by using NWP profiles from the surface up to the top of the atmosphere ? Both MWR and IRS have lower information on humidity compared to temperature so we could expect a larger benefit when using NWP information even below 4 km. Did you perform a sensitivity study using the whole NWP profile and not only the profile above 4 km ? Could you justify this choice to start at 4 km even for humidity ?

**Figure 5 :**

It seems that the configuration MWR + RAP degrades the vertical resolution of the configuration #1 with MWR only below 1.5 km : could you comment this result ? Do you have any explanation on

this slight degradation (which is overall pretty small compared to the large improvement that you get above 2 km) ?

**Figure 6 :**

The degradation due to the inclusion of the RAP data is significant for configuration #1 even below 4 km (from ~ 0.3 g/kg to 0.5 g/kg at 500m). This degradation is not really observed for the configuration #5 (IRS only). Do you have any hypothesis to explain this degradation when only the MWR is used ? I assume that this degradation could be due to the prior state covariance matrix used to spread the information from the observation level to adjacent vertical levels : considering that IRS has more information content in humidity than the MWR alone, the retrievals below 4 km might be better constrained by the observation while, in the MWR configuration, most of the prior state modification below 4 km is driven by the vertical correlations specified in the prior state covariance matrix. Did you test different prior state covariance matrices to evaluate the sensivity of the retrievals to this matrix ? Did you try to use the whole RAP profiles with its corresponding error covariance matrices to evaluate if this degradation is still observed ?

 **Figure 7 :**

The temperature bias is signifiantly increased in the configuration MWR + IRS compared to MWR or IRS only when averaged over 5km, I am puzzled by this result : could you include a discussion ?

**- Figure 8 :**

To be fair on your comment, I think the averaged bias and MAE of mixing ratio over 3 km should be presented as well as the inclusion of RAP significantly degrades the statistics of mixing ratio compared to MWR only below 3 km (which might give different results than your current conclusion that the impact of RAP only degrades slightly the bias and MAE).

 **Minor comments:**

- line 41 : aren't => are not.

- Table 1 : can you check the unit of mixing ratio (g / km ?)

- line 108 : isn't => is not

- line 368 : 0.5 g /km => 0.5 g/kg.

---

## Author Comment (AC1)

We thank the Referee for the thoughtful and detailed comments. We hope we have addressed all of the Referee's concerns and we think that our manuscript did benefit from the constructive comments made by all Referees. In the following text, the Referee's comments are in black and our answers are in red.

===========================================================================

**Review AMT-2023-263:**
**Sensitivity of thermodynamic profiles retrieved from ground-based**
**microwave and infrared observations to additional input data from active remote sensing**
**instruments and numerical weather prediction models**

**General comments:**

This paper evaluates the sensitivity of temperature and humidity retrievals to different combinations of ground-based remote sensing instruments as well as NWP model information. It uses the state of the art optimal estimation retrieval algorithm TROPOe that has been used in several scientific publications to demonstrate the improved accuracy of thermodynamic retrievals when combining together active remote sensing (water vapor lidar, RASS, ceilometers) and passive instruments (ground-based microwaver radiometers (MWR) and infrared spectrometers (IRS)). This new study is relevant for the scientific community due to the uniqueness of co-located remote sensing instruments at the same site (MWR, IRS, RASS, ceilometers and tower measurements) over a long time period (fall 2021-winter 2022). It also contributes to a new perspective of improvement for TROPOe by including NWP model information above 4 km altitude within the algorithm. The results show that RASS contributes to improved temperature retrievals within the boundary layer while the inclusion of NWP information significantly improves the temperature and humidity retrievals mainly above 3 km. Overall the manuscript is very well written, figures are well presented and explained.

We thank the Referee for the overall positive comments.

However, I am concerned by the conclusions of this study that are quite limited due to the small dataset of co-located radiosoundings available (only 15 RS). As clearly stated by the authors, conclusions taking into account cloudy-sky conditions could be different for the synergy between MWR and IRS observations for example.

We agree with the Referee's comment, as we stated several times in the text. We also considered one of the subsequent comments from the Referee and we think that our answer to that adds more perspective to the results that could derive from the combination of the passive remote sensors in case of presence of clouds.

I am also wondering if degradations in the retrievals could happen when including the NWP profile above 4 km directly in the observation vector (if I understood well) that might not been observed in this study due to the limited dataset of validation. In fact, the retrieval algorithm could try to minimize the distance between the atmospheric state and the a priori as well as observations with potentially non-consistent observations (the NWP profile could potentially try to push the algorithm in a direction while the observation in another direction). The author should clarify this point and better justify the methodology used.

This is a good point. If significant degradations were happening, we would notice it because the retrieval would struggle to find a valid solution. Fortunately, there are several variables included in the retrieval data files that give an indication of the quality of the retrieved thermodynamic profile that can be used as a robust quality control measure. To assure that the retrieval has the flexibility to consider the observations (if they diverge from the NWP) we have to specify the uncertainties for the NWP appropriately and not have them be too small. In our study, the uncertainty profiles for the RAP temperature and water vapor profiles are computed as the standard deviation over the surrounding neighboring grid points in the model. We additionally inflate the uncertainty of the RAP profiles by a factor of 3 for water vapor, while 1.5 °C is added to the temperature uncertainty. This has been now specified in the revised version of the manuscript in Section 2.5: "*The uncertainty profiles for the RAP temperature and water vapor profiles are computed as the standard deviation over the surrounding neighboring grid points in the model. We additionally inflate the uncertainty of the RAP profiles by a factor of 3 for water vapor, while 1.5 °C is added to the temperature uncertainty to assure that the retrieval has the flexibility to consider the observations (if they diverge from the NWP).*"

Additionally, I think a discussion on the differences in TROPOe results (DFS, vertical resolution, mean uncertainties, retrieved profiles) for clear-sky versus cloudy-sky days (as the observations are all available during one year) with respect to the different configurations could be beneficial to the paper. Even without available radio-soundings, DFS, vertical resolution and uncertainties could have been discussed as well statistical distributions of retrieval differences. If the TROPOe retrievals are available over a long time period and the authors can lead this analysis, I think it could improve the current manuscript.

We thank the Referee for this very smart suggestion. We had not thought of looking at the retrievals on cloudy days, because we didn't have radiosondes to compare the results with. Nevertheless, we recognize that even just looking at some of the TROPoe outputs, as suggested by the Referee, would provide insights of what results we could have in the presence of clouds. Unfortunately, we do not have observations available over one year. The dataset goes from the ~second half of September, 2021, to the ~end of January, 2022 (this is now clearly specified in the text, in the Introduction *"During fall 2021–winter 2022 (from the ~middle of September 2021 to the ~middle of January 2022)"*).
Moreover, the overall conditions over this time period were mostly clear skies.
Nevertheless, we used the several observations available and found 2 days that revealed the presence of clouds with corresponding relatively large values of liquid water path (LWP).
Specifically, these 2 days are 9/30/2021 and 1/9/2022.
In the figure below are presented time series of cloud base height from the ceilometer (CBH, in the upper panels) and liquid water path from the microwave radiometer (LWP, in the lower panels), for 9/30/3021 on the left, and for 1/9/2022 on the right.
In the bottom panels, all values of LWP are plotted in red in the time series, and values with LWP > 20 g kg$^{-1}$ are colored in blue.

[Figure]

For these 2 days we ran TROPoe using as input the MWR only, the IRS only, and the combination of the IRS + MWR. These configurations were named in the manuscript as configuration #1, #5, and #9, respectively. Time-height cross sections of temperature and mixing ratio for these TROPoe runs are presented in the next figures for 9/30/2021. CBH is plotted with the red asterisks.

[Figure]

[Figure]

[Figure]

[Figure]

*Time-height cross section of retrieved temperature for 30 September 2021 by the TROPoe run including the MWR and surface observations only (configuration #1)*

*Time-height cross section of retrieved temperature for 30 September 2021 by the TROPoe run including the IRS and surface observations only (configuration #5)*

[Figure]

[Figure]

*Time-height cross section of retrieved temperature for 30 September 2021 by the TROPoe run including MWR, IRS, and surface observations (configuration #9)*

Similarly, time-height cross sections of temperature and mixing ratio for these TROPoe runs are presented in the next figures for 1/9/2022. Again, CBH is plotted with the red asterisks.

[Figure]

Time-height cross section of retrieved temperature for 9 January 2022 by the TROPoe run including the MWR and surface observations only (configuration #1)

Time-height cross section of retrieved temperature for 9 January 2022 by the TROPoe run including the IRS and surface observations only (configuration #5)

[Figure]

Time-height cross section of retrieved temperature for 9 January 2022 by the TROPoe run including MWR, IRS, and surface observations (configuration #9)

From these examples, although we cannot assess which retrievals would agree better with an independent observation (i.e. radiosonde), we can notice the impact of the clouds in the retrievals relative to configuration #5 (IRS only) particularly, mostly in terms of time continuity above the cloud base height from previous profiles where the clouds were absent. For example, for 9/30/2021 we see a clear discontinuity around 0200 UTC for configuration #5, particularly in the mixing ratio plots, and at 1000, 1500, and 2000 UTC in the temperature retrievals of configuration #5, when we go from a situation with clouds observed between 1 and 1.5 km to clear-sky periods, or clouds being detected at a higher altitude. These discontinuities do not appear in the TROPoe retrievals of configuration #1 (MWR only), which was expected as stated in the manuscript, as clouds are markedly more opaque in the infrared than the microwave and provide little-to-no information above the cloud (whereas the MWR retrievals provide some sensitivity above the cloud). Also, as speculated in the manuscript, we notice that the retrievals of configuration #9 seems to benefit from the combination of MWR+IRS, as the information retrieved by the MWR supplements the lack of information above the cloud layer from the IRS. For configuration #9, the time discontinuities seen in the retrievals for configuration #5 are not present anymore.

For 1/9/2022, we also notice similar time discontinuity above the cloud base height in the retrievals of configuration #5, when going from clear-sky periods to cloud free ones (or with clouds at higher altitudes). Again, also for 1/9/2022, the time discontinuities seen in the retrievals of configuration #5 are not present anymore for configuration #9.

This seems promising of the combination of the MWR and IRS in cloudy conditions.

For these 2 days, we averaged cumulative degree of freedom for signal (DFS) and vertical resolution for temperature and mixing ratio, over the time periods with observed clouds and corresponding LWP > 20 g kg$^{-1}$.

In the following figure, these cumulative DFS as a function of the height for configurations #1, #5, and #9 are in the upper panels (temperature on the left and mixing ratio on the right) and vertical resolution as a function of the height for configurations #1, #5, and #9 are in the lower panels (temperature on the left and mixing ratio on the right).

From the upper left panel of the figure we notice that the cumulative DFS for temperature for #1 do not show much difference with the profile of configuration #1 presented in the manuscript and relative to clear-sky days. The cumulative DFS for temperature relative to configuration #5 do show a kink around 1 km agl, most likely due to the presence of clouds around that height. The cumulative DFS for temperature relative to configuration #9 show an increase in respect to those of configuration #1 and #5. This increase is larger to that showed in the manuscript and relative to configuration #9 in clear-sky days, which supports the speculation of the benefit that can be obtained by the combination of MWR and IRS in cloudy conditions.

From the upper right panel of the figure we notice that the cumulative DFS for mixing ratio for #1 again do not show much difference with the profile of configuration #1 presented in the manuscript and relative to clear-sky days. The cumulative DFS for mixing ratio relative to configuration #5 do show a large decrease compared to the profile of configuration #5 presented in the manuscript and relative to clear-sky days, most likely due to the presence of clouds. The cumulative DFS for mixing ratio relative to configuration #9 show an increase in respect to those of configuration #1 and #5. This increase again supports the speculation of the benefit that can be obtained by the combination of MWR and IRS in cloudy conditions.

From the lower left panel of the figure we notice a similar profile to that presented in the manuscript for configuration #1 (MWR only) for the vertical resolution of temperature, but a degradation in vertical resolution of temperature for the TROPoe configuration #5 (IRS only),

particularly in the lower part of the atmosphere, where clouds were present. Combining the MRW and IRS in the TROPoe runs, derives in an improvement of vertical resolution of temperature in the lower part of the atmosphere particularly, where the vertical resolution of configuration #5 was degraded.

From the lower right panel of the figure we notice a similar behavior in the profiles of vertical resolution of mixing ratio, i.e.: similar to those presented in the manuscript for configuration #1, a degradation of it for configuration #5 in the lower part of the atmosphere, and a benefit obtained by the combination of MWR and IRS (configuration #9).

[Figure]

*Upper panels: cumulative DFS for temperature (left) and mixing ratio (right) for the cloudy days 9/30/2021 and 1/9/2022, as a function of the height for TROPoe configurations #1, #5, and #9. Lower panels: vertical resolution of the retrieved TROPoe profiles (temperature on the left and mixing ratio on the right) for the same days, as a function of the height for the same TROPoe configurations.*

For an easier comparison of these results, in the following figure we present the same profiles as those included in the manuscript (computed for clear-sky days), removing the lines relative to the other configurations, not analyzed for this exercise.

[Figure]

Upper panels: cumulative DFS for temperature (left) and mixing ratio (right) as a function of the height for TROPoe configurations #1, #5, and #9, as included in the manuscript (clear-sky days). Lower panels: vertical resolution of the retrieved TROPoe profiles (temperature on the left and mixing ratio on the right) as a function of the height for the same TROPoe configurations, as included in the manuscript (clear-sky days).

Although this analysis is relative to only 2 cloudy days, the results motivate us even more to investigate the benefit of the combination of MWR and IRS during cloudy conditions on other datasets. As mentioned in the Conclusions of our manuscript, the instruments deployed at the Platteville, CO, site were later moved to other sites for other field campaigns, and the analysis performed at Platteville will be continued, when radiosonde launches will be available for comparison.

Additionally, the suggestion of the Referee motivated us to include an Appendix to the revised version of the manuscript that reports the results from the 2 cloudy days discussed in this response to the Referee. In the Appendix the two last figures are combined in one figure to allow for an easier comparison of the results in cloudy vs clear-sky days.

Finally, the manuscript does not clearly state if MWRs are used with off-zenith scans. Several publications have demonstrated the significant increase in DFS by including off-zenith observations. If MWR observations have been used with zenith only observations, the comparison with the AERI instrument is not really fair and we could except more accurate temperature and lapse rate retrievals with the inclusion of off-zenith channels. This will not drastically change the conclusion with RASS and RAP but it would definitely affect the conclusion comparing single passive instruments.

We thank the Referee for this comment. We certainly forgot to specify the scanning strategy used for the MWR observations.
For the Platteville dataset, the MWRs observed at the zenith and at 19.8° and 160.2° elevation angles on both sides of the zenith. We are aware that, when deployed in locations with unobstructed views, MWR's oblique scans can be performed down to 5° elevation angles and may provide better profile accuracy in the lowest 0–1 km agl layer (Crewell and Löhnert, 2007). Unfortunately, due to some obstructions, we could not go lower than 19.8° elevation angles. The scanning strategy has now been specified in Section 2.2 of the revised version of the manuscript: *"The MWR observed at the zenith and at 19.8° and 160.2° elevation angles on both sides of the zenith. We are aware that, when deployed in locations with unobstructed views, MWR's oblique scans can be performed down to 5° elevation angles and may provide better profile accuracy in the lowest 0–1 km agl layer (Crewell and Löhnert, 2007). Unfortunately, due to some obstructions, we could not go lower than 19.8° elevation angles."* The reference Crewell and Löhnert, 2007, has also been added to the Reference list.

After taking into account the scientific points listed, I would recommend the publication of this manuscript in AMT.

**Major comments :**

- **Section 2.2** : As shown in Djalalova 2022 , even after nitrogen calibration, significant biases can be observed in MWR observations. Optimal estimation is very sensitive to biases in observations. Did you implement any bias correction or quality control of the brightness temperatures before applying TROPOe ?

We indeed computed the biases in the brightness temperature MWR observations and corrected for it, before running TROPoe.
In order to compute the brightness temperature biases, we used the method referred to as "TROPoe BC" in Djalalova et al. (2022). We now specified this in the revised version of the manuscript, in Section 2.2: *"Additionally, in order to compute MWR's brightness temperature biases and correct for them before retrieving the thermodynamic profiles, we used the method referred to as "TROPoe BC" in Djalalova et al. (2022)."*

Even if the IRS is self-calibrating, was there any check on potential biases in IRS also ?

No additional checks on potential biases in the IRS were performed.

In line with the general evaluation : did you use off-zenith observations for the MWR ? If only zenith observations have been used, it should be clearly stated through all the manuscript that all conclusions comparing the MWR and IRS retrievals are underestimating the capability of current MWRs that are generally used with boundary layer scans to improve the vertical resolution of temperature profile. If another configuration with zenith and off-zenith observations could be included in the manuscript, it would be beneficial for the discussion.

As specified in the response to the Referee's earlier comment, the adopted scanning strategy for the MWR used for our study was to observe at the zenith and at 19.8 and 160.2 elevation angles on both sides of the zenith. The scanning strategy has now been specified in the revised version of the manuscript, in Section 2.2.

**Section 2.4** :It is not clear to me if the RAP model is used within the a priori profile or within the observation vector. If it is used within the observation vector how was defined the corresponding observation error covariance matrix ? As mentioned in the overal evaluation, I am concerned that this methodology could degrade the retrievals in case of larger errors in the NWP profile non consistent with the other observations that might not been observed due to the limited number of radiosounding observations. Could you clearify and justify the methodology ? Several publications using an alternative approach with NWP model used directly within the a priori profile should also be cited (Hewison 2007, Cimini et al 2015, Martinet et al 2020) to discuss the difference with your methodology.

We accept the Referee's suggestion and included reference to the other studies that use NWP models directly within the a priori profile.
In our study, though, dissimilarly from the studies mentioned by the Referee, we did not use the RAP as the a priori profile. We used the RAP as part of the observation vector. The uncertainty profiles for the RAP temperature and water vapor profiles are computed as the standard deviation over the surrounding neighboring grid points in the model. We additionally inflate the uncertainty of the RAP profiles by a factor of 3 for water vapor, while 1.5 °C is added to the temperature uncertainty. This has been now specified in the revised version of the manuscript in Section 2.5:
*"While other studies (Hewison 2007, Cimini et al 2015, Martinet et al 2020), employ an alternative approach with NWP model used directly within the a priori profile, in our study, we use the RAP as part of the observation vector. The uncertainty profiles for the RAP temperature and water vapor profiles are computed as the standard deviation over the surrounding neighboring grid points in the model. We additionally inflate the uncertainty of the RAP profiles by a factor of 3 for water vapor, while 1.5 °C is added to the temperature uncertainty to assure that the retrieval has the flexibility to consider the observations (if they diverge from the NWP)."*
In many studies we have been involved with, the information from NWP models has only been considered above 4 km agl (more reasoning on this choice can be found in the answer to the next Referees' question below). Research activities we are directing at, are rather focusing on the way to optimize the use of the prior being a key component of the TROPoe retrieval, to provide a better constraint on the ill-posed inversion problem.

**Figures 4 and 5:**

- When the information content from observations is small, the inclusion of external information from NWP models has a significant impact on the retrievals. This is demonstrated in this study in figures 4 and 5 both for temperature above 4 km agl and to a larger extent for humidity above 1.5 km. Could the improvement on water vapor be larger by using NWP profiles from the surface up to the top of the atmosphere ? Both MWR and IRS have lower information on humidity compared to temperature so we could expect a larger benefit when using NWP information even below 4 km. Did you perform a sensitivity study using the whole NWP profile and not only the profile above 4 km ? Could you justify this choice to start at 4 km even for humidity ?

We decided to include the NWP information only from 4 km up in the atmosphere because that is the height where both IRS and MWR's information content starts being very small. We decided to use only observations in the lower part of the atmosphere, inside the boundary layer, because we think these instruments (alone, or in combination) will be used for understanding physical processes happening in the boundary layer and assessing the ability of NWP models at reproducing them. Moreover, we believe that including the NWP input from above 4 km is less 'risky' as it will include information on the large-scale circulation (as the RAP assimilates observations collected by operational aircraft), while including NWP information in the lower part of the atmosphere might be 'riskier' for NWP with larger horizontal grid spacing or in sites of complex terrain. To assess the value of including NWP information in the lower part of the atmosphere we believe that a sensitivity study would need to be performed with NWP models with different horizontal grid spacing, at different sites, with different atmospheric conditions and also with different topographic characteristics. Our dataset does not seem appropriate for this task, but we agree that the Referee's comment is very good and will keep it in consideration when we'll have an appropriate dataset to investigate this possibility.

**Figure 5 :**

It seems that the configuration MWR + RAP degrades the vertical resolution of the configuration #1 with MWR only below 1.5 km : could you comment this result ? Do you have any explanation on this slight degradation (which is overall pretty small compared to the large improvement that you get above 2 km) ?

This is difficult to explain. We might speculate that the algorithm having to balance the information content from the observation in the lower part of the atmosphere with that from a NWP model in the upper part of it, might not always provide a smooth reconnection of the profile characteristics (i.e., in this case those of the vertical resolution).

**Figure 6 :**

The degradation due to the inclusion of the RAP data is significant for configuration #1 even below 4 km (from ~ 0.3 g/kg to 0.5 g/kg at 500m). This degradation is not really observed for the configuration #5 (IRS only). Do you have any hypothesis to explain this degradation when only the MWR is used ? I assume that this degradation could be due to the prior state covariance matrix used to spread the information from the observation level to adjacent vertical levels : considering that IRS has more information content in humidity than the MWR alone, the retrievals below 4 km might be better constrained by the observation while, in the MWR configuration, most of the prior state modification below 4 km is driven by the vertical correlations specified in the prior state

covariance matrix. Did you test different prior state covariance matrices to evaluate the sensivity of the retrievals to this matrix ? Did you try to use the whole RAP profiles with its corresponding error covariance matrices to evaluate if this degradation is still observed ?

As specified in a previous answer, we did not use the RAP as the a priori profile. We used the RAP as part of the observation vector. We agree with the Referee that the IRS has more information content in humidity than the MWR alone, so, for configuration #5 (IRS only) the retrievals below 4 km are better constrained by the observation. For configuration #1 (MWR only), differences between the NWP model and the observations above in the atmosphere (NWP model being drier than the MWR observations) might spread (to counterbalance) in the lower part of the atmosphere.

**Figure 7 :**
The temperature bias is signifiantly increased in the configuration MWR + IRS compared to MWR or IRS only when averaged over 5km, I am puzzled by this result : could you include a discussion ?

When both passive instruments have enough information the retrieval of the combination of the 2 has to balance between both inputs, which might not be optimal. Nevertheless, from Fig. 6 we see that the combination of the 2 passive instrument is beneficial in the lower part of the atmosphere, where the MWR tends to struggle identifying the correct height and shape of inversions (configuration #9 better than configuration #1). As a matter of fact, even in Fig. 7a we see that for the first point on the x axis the MAE of temperature is improved compared to the MWR only configuration, both when averaging over the lowest 5 km and lowest 3 km of the atmosphere.

**- Figure 8 :**
To be fair on your comment, I think the averaged bias and MAE of mixing ratio over 3 km should be presented as well as the inclusion of RAP significantly degrades the statistics of mixing ratio compared to MWR only below 3 km (which might give different results than your current conclusion that the impact of RAP only degrades slightly the bias and MAE).

As requested by the Referee, a new Fig.8 has been produced, now including the averages of Mixing Ratio MAE and Bias up to 3 km agl. Corresponding discussion has been revised in Section 4.2: *"the effect of the inclusion of the RAP in the TROPoe runs that only include the MWR as the passive instrument is to slightly degrade the MAE for mixing ratio in the lower in the 5 km of the atmosphere and a little more in the lowest 3 km, but it slightly improves the MAE values for the TROPoe runs that only include the IRS as the passive instrument when averaging in the lower in the 5 km of the atmosphere."*

**Minor comments:**
- line 41 : aren't => are not.
        Corrected as suggested.
- Table 1 : can you check the unit of mixing ratio (g / km ?)
        Units corrected, thanks for catching the mistake.
- line 108 : isn't => is not
        Corrected as suggested.
- line 368 : 0.5 g /km => 0.5 g/kg.

Units corrected, thanks for catching the mistake.

---

## Author Comment (AC2)

We thank the Referee for the thoughtful and detailed comments. We hope we have addressed all of the Referee's concerns and we think that our manuscript did benefit from the constructive comments made by all Referees. In the following text, the Referee's comments are in black and our answers are in red.

=================================================================

This is a well-written paper that investigates an important issue in ground-based remote sensing: what improvements arise when additional datastreams are included in the retrieval? For years now, TROPoe and its antecedent AERIoe have included RAP or RUC profiles in the calculations, but the improvement in the retrieval has not been fully quantified. This paper is of the appropriate scope and novelty for inclusion in AMT. I have a few corrections and suggestions that will help improve the readability and utility of this paper, but these should be easy to address. None of these issues rise beyond the level of minor corrections.

We thank the Referee for the overall positive comments.

Most significant issue:

Sound propagates differently at different times of day, and the uncertainties introduced into the RASS observations by horizontal winds are also going to have a strong diurnal cycle. Because of that, one might assume that the biases and MAEs exhibited here are not constant throughout the day, but instead have a noticeable diurnal cycle. It may be that there's an insufficient number of radiosondes, especially at non-daytime hours, to fully investigate this. Regardless, the possible diurnal impact should be discussed, even if it's not anticipated to be an important issue and can easily be dismissed.

We agree with the Referee that sounds propagate at different speeds as a function of the temperature. This is indeed the principle on which the RASS is based. We also agree that the horizontal winds have a diurnal cycle, which affects the possibility of the sound wave to be advected out of the measuring volume of the RASS, impacting the height coverage of the instrument. This was already specified in Section 2.2 *"Moreover, the maximum height reached by the RASS is variable and limited by the advection of the propagating sound wave out of the radar's field of view and by sound attenuation (a function of both radar frequency and atmospheric conditions such as temperature, humidity; May and Wilczak, 1993)"*. This is now reworded to *"Moreover, the maximum height reached by the RASS is variable and limited by the advection of the propagating sound wave out of the radar's field of view (which can be different at different times of the day, as horizontal winds can have a strong diurnal cycle) and by sound attenuation (a function of both radar frequency and atmospheric conditions such as temperature, humidity; May and Wilczak, 1993)"*. We are however not aware of a reference on RASS errors having a diurnal cycle and, as the Referee mentioned, our dataset would not allow for such investigation as we have a limited number of radiosondes and only at daytime hours.

Minor comments:

1. One small issue I had while reading this paper was understanding how the RASS was integrated into the retrieval. The TROPoe retrieval works in T and q space, and it's relatively easy to understand how the RAP profiles would be able to be included as they exist (or can easily be converted) into those variables. However, RASS is measuring Tv which is a unique variable for the retrieval. How does the retrieval address this? Is it just calculated at the end of each iteration from the interim T/q profile and compared to the RASS observations?

Yes, the Referee is correct and this is how it is done: before computing the Jacobian, the virtual temperature is computed from the state vector and compared to the RASS measured virtual temperature. This is now clarified in Section 3: *"We note that since the RASS measures virtual temperature, when this is included as input, the virtual temperature is computed at the end of each TROPoe iteration from the state vector and compared to the RASS measured virtual temperature."*

2. When I read line 76, noting that the radiosondes were interpolated to the TROPoe grid, I wondered why the sondes weren't smoothed with the retrieval averaging kernal instead. I see that was addressed later in Line 195. Still, I wonder if this is the best approach. Would it be better to smooth the sondes, calculate the differences as a function of height, and then interpolate the vertical profile of the differences to a common grid to facilitate the analysis between different combinations of instruments?

We agree with the Referee that smoothed radiosonde observed profiles can be computed using the averaging kernel. However, the averaging kernel depends on the retrieval parameters (e.g., which datasets are used as input in the TROPoe runs), so for our 12 configurations we would have 12 different averaging kernels. For each of these the smoothed radiosonde profile can be quite different from each other and also from the original unsmoothed radiosonde profile. Consequently, while comparison of the retrievals to the relative averaging kernels radiosonde profiles can be used to minimize the vertical representativeness effects due to the different vertical resolutions of these profiles, we were not convinced that a statistical comparison between the 12 TROPoe configurations would be fair if each of their retrieved profiles is compared to a different averaging kernel-smoothed radiosonde profile. Therefore, we decided to present the statistical analysis in the manuscript comparing the various TROPoe retrieval configurations to the unsmoothed radiosonde profiles, just interpolated to the same vertical levels of the retrieved profiles. Additionally, we decided that we want to ultimately present statistical values relative to real radiosonde observations.

3. In certain instances, the additional observation vector entries degrade the profiles, either by increased bias or MAE. This is an interesting finding that ought to be discussed more. What is causing this, how consistent is it (are a couple of retrievals dragging everything down, or are most of the retrievals behaving similarly)?

When both passive instruments have enough information the retrieval of the combination of the 2 has to balance between both inputs, which might not be optimal. Additionally, physical retrievals might struggle to overcome systematic biases in the observations, and although we are bias correcting the MWR observations there may still be some residual bias that isn't accounted for correctly. Nevertheless, from Fig. 6 we see that the combination of the 2

passive instruments is beneficial in the lower part of the atmosphere (< 2 km), where the MWR tends to struggle identifying the correct height and shape of inversions (configuration #9 better than configuration #1). As a matter of fact, even in Fig. 7a we see that for the first point on the x axis the MAE of temperature is improved compared to the MWR only configuration, both when averaging over the lowest 5 km and lowest 3 km of the atmosphere.

Technical corrections:

Line 42: this sentence is somewhat awkwardly phrased and would be easier to interpret if rewritten.

The above-mentioned sentence has been re-worded in the revised version of the manuscript from: *"Several ground-based sensors are nowadays available and active in many geographical locations including in-situ or remote, and active or passive sensors."*, to: *"Several ground-based sensors (including in-situ or remote, and active or passive sensors) are currently available and operational in many geographical locations."*

Line 43: I don't think it's quite correct to state that in situ sensors only provide point measurements. Surface meteorology stations like ASOS only provide observations at a single point fixed in 3D space, that is true. However, aircraft-based observations like AMDAR are in situ but producing moderately-dense vertical profiles and the sheer density of these observations next to major airports produces a multidimensional web of observations.

We agree with the Referees, and mention of the aircraft-based observations has been included in the revised version of the manuscript, in the Introduction: *"In-situ sensors only provide point measurements (except for aircraft-based observations that can produce moderately-dense vertical profiles, although only sporadically in time)"*.

Line 54 and elsewhere: the name of the radiosonde is the Vaisala RS-41, not Vaisala-41.

Corrected, thanks for catching the mistake.

Line 96: it is somewhat unclear what the upper range of the IRS is: why does it range from 3000-5000 cm^-1? Did the two IRSes each have a different spectral range?

The IRS used in our study goes out to 5,000 $cm^{-1}$, but the AERI (which reference we used) measures out to 3,000 $cm^{-1}$.

Line 440: necessarily, not necessary

Corrected, thanks for catching the mistake.

Line 444: cloudy conditions, not cloudy-conditions.

Corrected, thanks for catching the mistake.

Line 454: Based on my experience as an AMT author, making data available by request may be insufficient for their standards. It may be good to identify an archive where these data may be stored and given a DOI.

Following the Referee's suggestion, we uploaded all radio soundings and TROPoe output files in the Zenodo data archive: 10.5281/zenodo.10815373.

---

## Author Response (AR3)

**Answers to the Editor, Referee #1, and Referee #2**

We thank the Editor and the Referees for the positive comments to our revised manuscript. In the following text, the Editor's and Referees' comments are in black and our answers are in red.

Editor's comment:
Dear authors,
based on the referees' reports and my own judgment, I believe the manuscript can be accepted for publication on AMT subject to the minor modifications suggested by one of the referees (Report #2).
I also encourage the authors to link their paper to the AMT/GMD PROBE special issue [1] to possibly increase the visibility in the BL profiling community. If agreed, just ask the editorial staff to be linked.
Congratulations!
[1] https://amt.copernicus.org/articles/special_issue1209.html

We thank the Editor for the comments and suggestions. We have taken Referee's #1 comments in consideration and have included the suggested modifications to our manuscript. Additionally, we'll be happy to link our paper to the AMT/GMD PROBE special issue.

==============================

Referee#1 comment:
First of all, I would like to thank the authors for their efforts to take into account most of the suggestions and propose this new improved version of the manuscript. I only suggest minor corrections and I recommend publication to AMT.

Thanks for your comments and suggestions. We have taken your comments into consideration and have included the suggested modifications to our manuscript.

Minor corrections :
→ Appendix :
- line 484 : it's not => it is not

Changed as suggested.

- I think this appendix is really great to demonstrate the benefit of using the sensor synergy of infrared and microwave frequencies for cloud-sky applications.
I think it would be really valuable to include the Appendix directly within the paper.
But I let the authors decide with the editor if they really prefer to keep it in the appendix.

We agree with the Referee that the Appendix brings valuable insights on the benefit of using the sensor synergy of infrared and microwave frequencies for cloud-sky applications. We again thank the Referee for suggesting to investigate cloudy days in the first round of reviews. We would rather prefer to keep it in the Appendix, as a preliminary teaser on future research investigations.

→ Figure 6 : for the degradation of humidity with MWR + RAP instead of MWR : even if we cannot be entirely sure, I think it would be nice to include your explanation within the paper:
IRS has more information content in humidity than the MWR alone, so, for configuration #5 (IRS only) the retrievals below 4 km are better constrained by the observation. For configuration #1 (MWR only), differences between the NWP model and the observations above in the atmosphere (NWP model being drier than the MWR observations) might spread (to counterbalance) in the lower part of the atmosphere.

We thank the Referee for the suggestion. Some explanatory text text has been added to the revised version of the manuscript, in Section 4.2.

→ Figure 7 : could you just describe within the paper the fact that the bias is degraded for the configuration IRS + MWR compared to MWR (even if we do not know why but just to mention it in the text.

The suggested comment has been added to the revised version of the manuscript, in Section 4.2.

==============================

Referee#2 comment:
I am satisfied with the authors' responses to my concerns and I believe this paper should be accepted.

We thank the Referee for the positive comments to our revised manuscript.